# Transcriptional Landscape of Ectomycorrhizal Fungi and Their Host Provides Insight into N Uptake from Forest Soil

Carmen Alicia Rivera Pérez,[a] Dennis Janz,[a] Dominik Schneider,[b,c] Rolf Daniel,[b,c] Andrea Polle[a]

[a]Forest Botany and Tree Physiology, Büsgen Institute, Georg-August University of Göttingen, Göttingen, Germany
[b]Department of Genomic and Applied Microbiology, Institute of Microbiology and Genetics, Georg-August University of Göttingen, Göttingen, Germany
[c]Göttingen Genomics Laboratory, Institute of Microbiology and Genetics, Georg-August University of Göttingen, Göttingen, Germany

**ABSTRACT** Mineral nitrogen (N) is a major nutrient showing strong fluctuations in the environment due to anthropogenic activities. The acquisition and translocation of N to forest trees are achieved mainly by highly diverse ectomycorrhizal fungi (EMF) living in symbioses with their host roots. Here, we examined colonized root tips to characterize the entire root-associated fungal community by DNA metabarcoding-Illumina sequencing of the fungal internal transcribed spacer 2 (ITS2) molecular marker and used RNA sequencing to target metabolically active fungi and the plant transcriptome after N application. The study was conducted with beech (*Fagus sylvatica* L.), a dominant tree species in central Europe, grown in native forest soil. We demonstrate strong enrichment of $^{15}N$ from nitrate or ammonium in the ectomycorrhizal roots by stable-isotope labeling. The relative abundance of the EMF members in the fungal community was correlated with their transcriptional abundances. The fungal metatranscriptome covered Kyoto Encyclopedia of Genes and Genomes (KEGG) and Eukaryotic Orthologous Groups (KOG) categories similar to those of model fungi and did not reveal significant changes related to N metabolization but revealed species-specific transcription patterns, supporting trait stability. In contrast to the resistance of the fungal metatranscriptome, the transcriptome of the host exhibited dedicated nitrate- or ammonium-responsive changes with the upregulation of transporters and enzymes required for nitrate reduction and a drastic enhancement of glutamine synthetase transcript levels, indicating the channeling of ammonium into the pathway for plant protein biosynthesis. Our results support that naturally assembled fungal communities living in association with the tree roots buffer nutritional signals in their own metabolism but do not shield plants from high environmental N levels.

**IMPORTANCE** Although EMF are well known for their role in supporting tree N nutrition, the molecular mechanisms underlying N flux from the soil solution into the host through the ectomycorrhizal pathway remain widely unknown. Furthermore, ammonium and nitrate availability in the soil solution is subject to frequent oscillations that create a dynamic environment for the tree roots and associated microbes during N acquisition. Therefore, it is important to understand how root-associated mycobiomes and the tree roots handle these fluctuations. We studied the responses of the symbiotic partners by screening their transcriptomes after a sudden environmental flux of nitrate or ammonium. We show that the fungi and the host respond asynchronously, with the fungi displaying resistance to increased nitrate or ammonium and the host dynamically metabolizing the supplied N sources. This study provides insights into the molecular mechanisms of the symbiotic partners operating under N enrichment in a multidimensional symbiotic system.

**KEYWORDS** ammonium, *Fagus sylvatica*, fungi, metatranscriptome, mycorrhiza, nitrate, nitrogen stress, symbiosis

Address correspondence to Carmen Alicia Rivera Pérez, rivera@gwdg.de.

The authors declare no conflict of interest.

Soil N availability is generally a main limiting factor for primary productivity across terrestrial ecosystems, including temperate forests (1, 2). In forest soil, soluble mineral N pools consist of nitrate and ammonium, whose quantities fluctuate in time and space, depending on the soil properties, meteorological conditions, anthropogenic N inputs, and biological processes such as mineralization, immobilization, and denitrification (3–12). While nitrate ions are highly mobile in soil solution and easily lost by leaching, ammonium cations are generally bound to soil colloids and retained in topsoil (13, 14). Consequently, mineral N nutrition of plants and microbes must cope with dynamic N availabilities in the environment.

The mutualistic association of certain species of soil ectomycorrhizal fungi (EMF) with the root tips of forest trees is an ecological advantage to support the nutrition of the host from various environmental N sources (15–20). The vast majority of the root systems of individual trees in temperate forests are naturally colonized by a diverse spectrum of EMF species forming compound organs known as ectomycorrhizas and variably composed fungal communities (21–24). These ectomycorrhizas consist of root and fungal cells that mediate bidirectional nutrient exchange. EMF acquire N from the environment, transfer it to the root, and receive host-derived carbon in return (25, 26). EMF show strong interspecific differences in N acquisition (27, 28). Early laboratory experiments showed that when the mycelium of EMF colonizing the roots of *Pinus sylvestris* and *Fagus sylvatica* was supplied with either ammonium or nitrate, the N sources became predominantly incorporated into the amino acids glutamate, glutamine, aspartate, asparagine, and alanine (29, 30). When ammonium and nitrate were supplied at equimolar concentrations to the mycelium of *Paxillus involutus*, ammonium incorporation into amino acids occurred in the fungus, and nitrate remained almost unchanged, suggesting that EMF assimilate ammonium more readily than nitrate into amino acids prior to delivering it to the plant (31). In general, EMF have a preference for ammonium in comparison to nitrate (32, 33), but their ability to metabolize nitrate is also widespread (34, 35). Silencing of the nitrate reductase gene (*NR*) in *Laccaria bicolor* impaired the formation of mycorrhizas with poplar (36), implying an important role of EMF in nitrate acquisition for the host.

The process of N transfer to the host through the mycorrhizal pathway starts at the soil-fungus interface, where different N forms are taken up from the soil solution by fungal membrane transporters; N is then translocated through the fungal mantle, which enwraps the root tip, into the intraradical hyphae and finally exported to the symbiotic interface, becoming available for the plant (37–42). Studies on *Amanita muscaria*, *Hebeloma cylindrosporum*, *Laccaria bicolor*, and *Tuber melanosporum* have led to the hypotheses that ammonium is exported from the intraradical hyphae to the symbiotic interface through ammonia/ammonium transport out (Ato) proteins, voltage-dependent cation channels, and aquaporins (37, 43–46) and that amino acid export could occur through acid quinidine resistance 1 proteins in *Laccaria bicolor* and *Hebeloma cylindrosporum* (38, 44, 47). Moreover, the EMF-mediated supply of ammonium and nitrate to the roots is supported by the upregulation of the ammonium transporter (*AMT*) (43) and nitrate transporter (*NRT*) genes in ectomycorrhizal poplar roots, like *PttNRT2.4A* with *Amanita muscaria* (48) and *PcNRT1.1* and *PcNRT2.1* with *Paxillus involutus* (49).

Once nitrate is taken up by NRTs, it is intracellularly reduced to nitrite by NR and then to ammonium by nitrite reductase (NiR), and ammonium is ultimately incorporated into glutamine and glutamate (47, 50, 51) through the cyclic operation of glutamine synthetase (GS) and glutamate synthase (GOGAT). GS catalyzes the formation of glutamine by the transfer of ammonium to glutamate, and GOGAT then transfers the amino group from glutamine to 2-oxoglutarate, generating two molecules of glutamate, whereas in the alternative pathway, the enzyme glutamate dehydrogenase (GDH) catalyzes the reductive amination of one molecule of 2-oxoglutarate using ammonium to generate one molecule of glutamate (50, 51). Both the GS/GOGAT and GDH pathways operate in EMF, but variations are common among species or symbiotic

systems depending on the plant and fungal partners (52–54). In contrast to EMF, in plants, the GS/GOGAT pathway predominates, and GDH plays a minor role in ammonium incorporation into organic N forms (55). Currently, the molecular processes used by EMF for supplying mineral N to the host under field conditions are unknown. Uncovering these molecular activities will enable a better understanding of tree N nutrition and N cycling in the ecosystem.

Despite the well-recognized importance of the mycorrhizal pathway as a relevant route whereby tree roots acquire N, knowledge of the molecular mechanisms operating in the uptake, transport, and delivery of N to the host is limited to a few model EMF. It is also unknown how EMF and the colonized root cells respond to variation in mineral N availabilities. The 1000 Fungal Genomes Project (56) along with the *Fagus sylvatica* genome (57) provide a platform for disentangling fungal and plant transcription profiles in natural communities engaged in active symbioses. We took advantage of new tools to unravel these responses in natural forest soil by administering an N dose corresponding to 29 kg N ha$^{-1}$ year$^{-1}$, a quantity in the range of an N-saturated beech forest (58, 59). To control N uptake and to distinguish the responses to different N forms, we fertilized with either $^{15}$N-labeled ammonium or $^{15}$N-labeled nitrate and then studied the transcriptional responses separately for EMF and the host trees using ectomycorrhizal root tips (EMRTs). We used DNA barcoding to describe the composition of the root-associated fungal community and RNA sequencing (RNA-seq) to capture the metabolically active fungi associated with roots. We hypothesized that (i) the fungal community structure is unaffected after short-term exposure to elevated N and (ii) the transcriptional responses of metabolically active EMF reveal molecular activities related to the uptake and assimilation of nitrate and ammonium. Since nitrate assimilation requires a series of reduction steps to ammonium before its incorporation into amino acids, both distinct and overlapping responses to nitrate and ammonium availability were expected to be imprinted in the transcription profiles of the symbiotic partners. Furthermore, we hypothesized that (iii) EMF buffer environmental fluctuations in N for the plant resulting in strong N-induced responses in the fungal metatranscriptome but only marginal effects on the root transcriptome or (iv) the entire symbiotic system forms a "holobiont" where the host and the EMF partners display synchronized and similar N responses.

## RESULTS

**Abundance of root-associated fungal genera corresponds to transcriptional abundance.** The global fungal community associated with beech roots in this experiment was dominated by six genera containing ectomycorrhizal fungi (*Amanita* [7.18%], *Cenococcum* [9.05%], *Scleroderma* [4.83%], and *Xerocomus* [29.17%]), ericoid fungi (*Oidiodendron* [1.09%]), and saprotrophic fungi (*Mycena* [3.75%]) (Fig. 1A; see also Data Set S1 at Dryad [132]). The remaining taxa were rare (<1% per genus) and belonged to the phyla Ascomycota (2.31%), Basidiomycota (2.51%), Mucoromycota (0.11%), and Mortierellomycota (0.02%), and the rest were fungi of unknown phylogenetic lineages (39.98%) (Fig. 1A; see also Data Set S1 at Dryad [132]). We did not detect any significant effects of short-term ammonium or nitrate treatment on fungal operational taxonomic unit (OTU) richness ($F_{2,9}$ = 0.288; $P$ = 0.756), Shannon diversity ($F_{2,9}$ = 0.437; $P$ = 0.659) (see Table S1 in the supplemental material), or the composition of the fungal OTU assemblages ($R^2$ = 0.146; pseudo-$F_{2,9}$ = 0.767; $P$ = 0.861; 9,999 permutations [adonis]) (Fig. S1A). We aggregated the RNA counts of the fungi belonging to the same genus (Fig. 1B). The transcript abundances obtained for individual genera were variable within replicates and treatment groups. However, there were no significant differences at the fungal metatranscriptome level in the nitrate, ammonium, or control treatments ($R^2$ = 0.198; pseudo-$F_{2,9}$ = 1.110; $P$ = 0.353; 9,999 permutations [adonis]) (Fig. 1B; Fig. S1B). The internal transcribed spacer 2 (ITS2) gene relative abundances for the different fungal genera and the transcript abundances mapped to the specific reference fungal species for each sample according to treatment are shown in Fig. S2.

The transcript abundance of a specific fungal genus was strongly correlated with

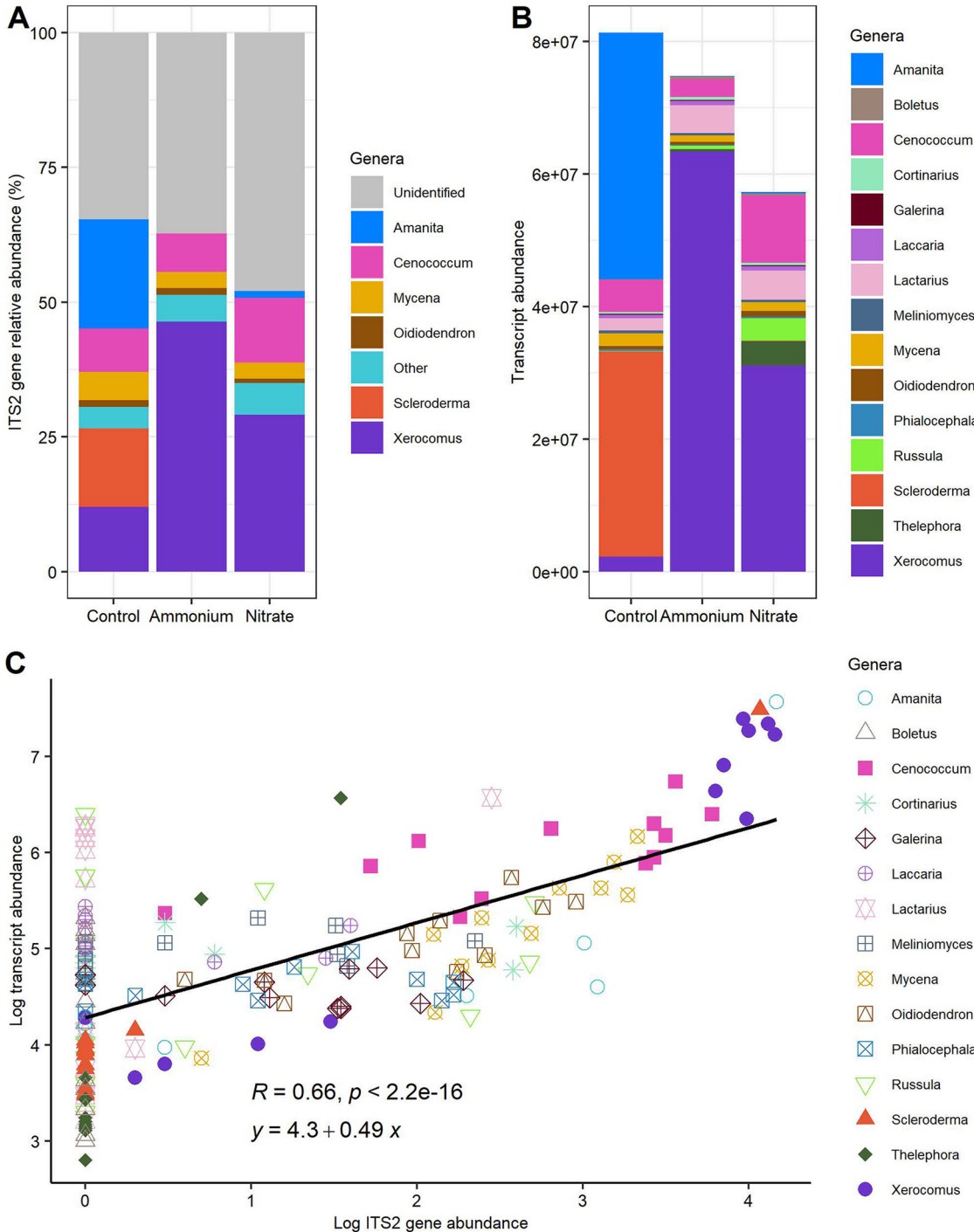

**FIG 1** Relative abundance of root-associated fungi (RAF) based on ITS2 barcoding (A), raw counts of the metabolically active fungi based on RNA sequencing characterized by taxonomy (B), and Pearson correlations between DNA-based and RNA-based abundances of the fungal genera (C). RAF were studied on roots of European beech (*Fagus sylvatica*) grown in native forest soil and treated with either water (control), ammonium, or nitrate for 2 days before harvest (*n* = 4 per treatment).

the ITS2-based abundance of that same genus (*R* = 0.66; *P* < 0.001 [Pearson]) (Fig. 1C), supporting that the molecular-level activities of abundant and metabolically active fungi associated with the beech roots were captured. Fungi with low abundances as determined by ITS2-based metabarcoding also showed significant transcript abundances (Fig. 1C), implying that low-abundance fungi may still contribute significantly to the molecular activities of the root mycobiome.

**Fungal metatranscriptomes cover fungal metabolism, which hardly responds to N treatments.** The RNA data containing ectomycorrhizal, ericoid mycorrhiza, endophytic, and saprotrophic fungi comprised a total of 175,531 transcript identifiers or gene models, covering 3,759 unique Eukaryotic Orthologous Groups of protein identifiers (KOGs). From these, 122,437 transcript identifiers (covering 3,708 unique KOGs) belong purely to the EMF (see Data Set S2 at Dryad [132]). Given the patchy occurrence of fungi within replicates (Fig. S2), the fungi were aggregated according to KOGs into a metatranscriptome, and after normalization in DESeq2, the full-list fungal metatranscriptome (17 fungi) (Table 1) resulted in 3,619 unique KOGs, whereas the EMF-specific metatranscriptome (13 EMF species) (Table 1) comprised 3,593 KOGs (see Data Set S3 at Dryad [132]). We evaluated the molecular functions of the EMF metatranscriptome according to KOG functional classifications. All 25 KOG functions were represented and categorized into "cellular processing and signaling" (1,159 KOGs), "information, storage, and processing" (956 KOGs), "metabolism" (796 KOGs), "poorly characterized" (817 KOGs), and multiple function assignments (135 KOGs) (Fig. 2). The frequencies of these functional classifications roughly reflected the same pattern of KOG frequencies present *in silico* in the model EMF *Laccaria bicolor* and that of *Laccaria* sp. on the beech roots (Fig. 2).

We further tested with DESeq2 whether the KOGs belonging to the full fungal metatranscriptome list or only to the EMF metatranscriptome were significantly differentially expressed in response to ammonium or nitrate treatment relative to the control. In response to ammonium, not a single KOG was significantly affected (see Data Set S3 at Dryad [132]). In response to nitrate, one differentially expressed KOG (KOG4381) was detected in the full fungal metatranscriptome list, and two KOGs (KOG4381 and KOG4431) were detected in the EMF metatranscriptome (see Data Set S3 at Dryad [132]). KOG4381 (RUN domain-containing protein) was upregulated in both the full fungal metatranscriptome list ($\log_2$ fold change = 9.175; false discovery rate [FDR]-adjusted $P = 0.024$) and the EMF metatranscriptome ($\log_2$ fold change = 9.100; FDR-adjusted $P = 0.021$) (see Data Set S3 at Dryad [132]). The function of KOG4381 is "signal transduction mechanisms" under the "cellular processes and signaling" category. Conversely, KOG4431 (uncharacterized protein induced by hypoxia) has "poorly characterized function" and was downregulated ($\log_2$ fold change = $-1.180$; FDR-adjusted $P = 0.021$) in response to nitrate in the EMF metatranscriptome (see Data Set S3 at Dryad [132]). Some fungi (e.g., *Cenococcum geophilum* and *Xerocomus badius*) occurred in almost all samples (Fig. S2), but because of overall low transcriptome coverage, we did not test differential responses to N treatments in individual fungi.

Mapping the EMF metatranscriptome to the Kyoto Encyclopedia of Genes and Genomes (KEGG) pathway database with *Laccaria bicolor* as the reference revealed 108 metabolic pathways, including "biosynthesis of amino acids," "carbon metabolism," and "nitrogen metabolism" (Table S2). From a total of 952 unique Enzyme Commission (EC) numbers, the complete ones (866) were mapped, and the partial ones (86) were excluded to avoid inaccurate multiple reaction assignments (60). KEGG pathway enrichment analysis pooling all treatments revealed putative metabolic functions of the EMF metatranscriptome with 11 significantly enriched pathways (FDR-adjusted $P <$ 0.05), mainly for energy, carbon, and amino acid metabolism: "glycolysis/glucogenesis," "pentose phosphate pathway," "pyruvate metabolism," "amino sugar and nucleotide sugar metabolism," "pyrimidine metabolism," "biosynthesis of amino acids," and "arginine biosynthesis" (Table 2). "Nitrogen metabolism," represented by the enzymes GS (EC 6.3.1.2), GDH (EC 1.4.1.2), nitrilase (EC 3.5.5.1), and carbonic anhydrase (EC 4.2.1.1), was covered but not significantly enriched (FDR-adjusted $P = 0.059$). KEGG pathway enrichment analysis of the full fungal metatranscriptome list (920 unique and complete Enzyme Commission numbers) also returned similar results, with 11 significantly enriched pathways (Table 2) and "nitrogen metabolism" not significantly enriched (FDR-adjusted $P = 0.057$). After manually searching the complete fungal metatranscriptomic database, including ectomycorrhizal and nonectomycorrhizal fungi (see Data Set S2 at Dryad [132]), transcripts encoding proteins and enzymes required for fungal N uptake and assimilation

**TABLE 1** Taxonomy of the genera representing the beech root-associated fungal community and the reference species chosen from the JGI MycoCosm database for mapping the RNA sequencing data

| Phylum | Order | Genus | Species | Trophic mode | Guild(s)[a] | JGI short name | JGI name | JGI reference(s) |
|---|---|---|---|---|---|---|---|---|
| Ascomycota | Helotiales | Phialocephala | Phialocephala scopiformis | Symbiotroph | Endophyte | Phisc1 | Phialocephala scopiformis 5WS22E1 v1.0 | 133 |
| Ascomycota | Helotiales | Oidiodendron | Oidiodendron maius | Symbiotroph | Ericoid mycorrhiza | Oidma1 | Oidiodendron maius Zn v1.0 | 134, 135 |
| Ascomycota | Helotiales | Meliniomyces | Meliniomyces bicolor | Symbiotroph | Ectomycorrhiza and ericoid mycorrhiza | Melbi2 | Meliniomyces bicolor E v2.0 | 135 |
| Ascomycota | Mytilinidales | Cenococcum | Cenococcum geophilum | Symbiotroph | Ectomycorrhiza | Cenge3 | Cenococcum geophilum 1.58 v2.0 | 82 |
| Basidiomycota | Agaricales | Galerina | Galerina marginata | Saprotroph | Saprotroph | Galma1 | Galerina marginata v1.0 | 136 |
| Basidiomycota | Agaricales | Mycena | Mycena galopus | Saprotroph | Leaf litter decomposer | Mycgal1 | Mycena galopus ATCC-62051 v1.0 | 137 |
| Basidiomycota | Agaricales | Amanita | Amanita muscaria | Symbiotroph | Ectomycorrhiza | Amamu1 | Amanita muscaria Koide v1.0 | 134 |
| Basidiomycota | Agaricales | Amanita | Amanita rubescens | Symbiotroph | Ectomycorrhiza | Amarub1 | Amanita rubescens Príilba v1.0 | 137 |
| Basidiomycota | Agaricales | Cortinarius | Cortinarius glaucopus | Symbiotroph | Ectomycorrhiza | Corgl3 | Cortinarius glaucopus AT 2004 276 v2.0 | 137 |
| Basidiomycota | Agaricales | Laccaria | Laccaria amethystina | Symbiotroph | Ectomycorrhiza | Lacam2 | Laccaria amethystina LaAM-08-1 v2.0 | 134 |
| Basidiomycota | Agaricales | Laccaria | Laccaria bicolor | Symbiotroph | Ectomycorrhiza | Lacbi2 | Laccaria bicolor v2.0 | 138 |
| Basidiomycota | Boletales | Imleria | Imleria badia (syn., Xerocomus badius) | Symbiotroph | Ectomycorrhiza (saprobic abilities) | Xerba1 | Xerocomus badius 84.06 v1.0 | 137 |
| Basidiomycota | Boletales | Boletus | Boletus edulis | Symbiotroph | Ectomycorrhiza | Boledp1 | Boletus edulis Príilba v1.0 | 137 |
| Basidiomycota | Boletales | Scleroderma | Scleroderma citrinum | Symbiotroph | Ectomycorrhiza | Sclci1 | Scleroderma citrinum Foug A v1.0 | 134 |
| Basidiomycota | Russulales | Russula | Russula ochroleuca | Symbiotroph | Ectomycorrhiza | Rusoch1 | Russula ochroleuca Príilba v1.0 | 137 |
| Basidiomycota | Russulales | Lactarius | Lactarius quietus | Symbiotroph | Ectomycorrhiza | Lacqui1 | Lactarius quietus S23C v1.0 | 137 |
| Basidiomycota | Thelephorales | Thelephora | Thelephora terrestris | Symbiotroph | Ectomycorrhiza | Theter1 | Thelephora terrestris UH-Tt-Lm1 v1.0 | 137 |

[a]Guild is the type of known functional group for the species used as a reference for mapping the RNA sequence read data.

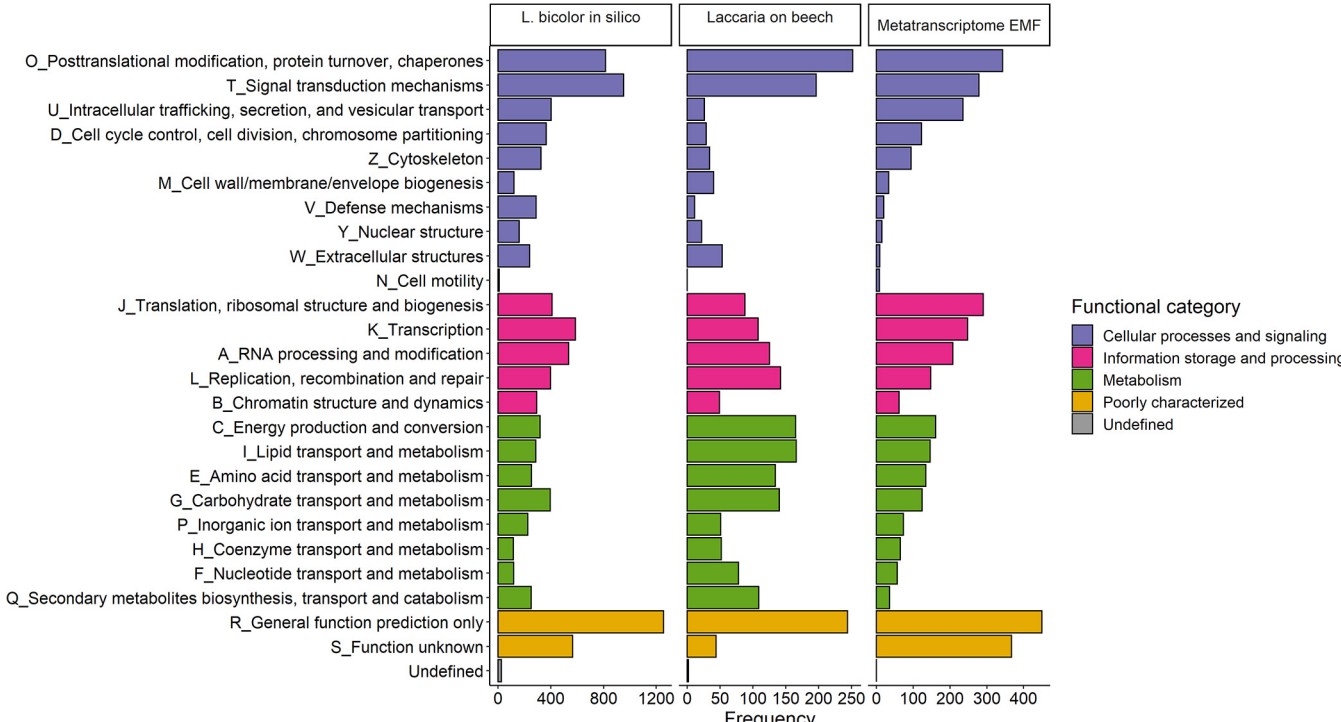

**FIG 2** Functional classification of the metatranscriptome of the ectomycorrhizal fungi (EMF) according to KOG functional groups. The distribution of KOG functions for the model ectomycorrhizal fungus *Laccaria bicolor* (*in silico* analyses of the published genome [138]) and KOG functions in the transcriptome of the genus *Laccaria* in this experiment (*Laccaria* on beech) and the entire ectomycorrhizal fungal metatranscriptome in this experiment (Metatranscriptome EMF) are shown.

were discovered. These clustered according to the fungal species instead of putative transporter/enzyme function (Fig. 3). The samples did not clearly cluster according to treatments but formed two main clusters, one containing the majority of nitrate- and ammonium-treated samples (6/8), and the other containing the majority of the controls (3/4). However, these differences were not significant ($R^2 = 0.176$; pseudo-$F_{2,9} = 0.96161$; $P = 0.475$ [adonis]).

**$^{15}N$ application records strong N uptake by roots with increased root N concentrations.** The EMRTs showed strong $^{15}N$ enrichment in response to $^{15}NH_4^+$ and $^{15}NO_3^-$ treatment (Table 3), although specific effects related to mineral N provision were not discovered in the EMF metatranscriptome. The $^{15}N$ enrichment in the root system decreased with increasing distance from the root tips and was about 2 times lower in fine roots and about 6 to 8 times lower in coarse roots than in EMRTs (Table 3). The total N content of the $^{15}N$-treated roots was slightly and significantly increased in comparison to control roots (Table 3), supporting that short-term N application caused enhanced N uptake. Thus, the N treatments triggered a significant decrease in the fine root C/N ratio compared to the controls (Table 3). The soil N content was not markedly affected by $^{15}N$ application, and the $^{15}N$ signatures of nitrate- and ammonium-treated soils did not differ from each other (Table 3). Overall, the beech root systems accumulated 1.5% ± 0.7% and 1.2% ± 0.6% of $^{15}N$ from ammonium and nitrate, respectively (Table 3). Since the assimilation of inorganic nitrogen requires carbon skeletons (51), we measured fine root nonstructural carbohydrate concentrations. However, no significant effects of N treatment on the carbohydrate concentrations were detected (Table 3).

**The beech transcriptome responds to nitrate and ammonium treatments activating N assimilation.** Mapping of the RNA reads to the beech genome resulted in a total of 55,408 beech transcript identifiers or gene models before normalization (see Data Set S4 at Dryad [132]) and 27,135 beech gene models after normalization (see Data Set S5 at Dryad [132]) in DESeq2. Ammonium and nitrate treatments resulted

**TABLE 2** KEGG pathway enrichment analysis of the ectomycorrhizal fungal and full fungal metatranscriptomes[a]

| Term | Term description | P value | |
|------|------------------|---------|------|
| | | EMF | All fungi |
| KEGG:01100 | Metabolic pathways | 6.1E−17 | 3.4E−17 |
| KEGG:01110 | Biosynthesis of secondary metabolites | 8.5E−11 | 1.5E−10 |
| KEGG:00230 | Purine metabolism | 7.8E−07 | 8.8E−07 |
| KEGG:01240 | Biosynthesis of cofactors | 7.1E−05 | 8.3E−05 |
| KEGG:01230 | Biosynthesis of amino acids | 7.1E−05 | 8.3E−05 |
| KEGG:01200 | Carbon metabolism | 1.0E−04 | 1.2E−04 |
| KEGG:00010 | Glycolysis/gluconeogenesis | 1.8E−03 | 2.0E−03 |
| KEGG:00520 | Amino sugar and nucleotide sugar metabolism | 2.4E−02 | 2.5E−02 |
| KEGG:00030 | Pentose phosphate pathway | 2.5E−02 | 2.6E−02 |
| KEGG:00220 | Arginine biosynthesis | 2.9E−02 | 3.0E−02 |
| KEGG:00240 | Pyrimidine metabolism | 2.9E−02 | 3.0E−02 |
| KEGG:00680 | Methane metabolism | 5.9E−02 | 5.8E−02 |
| KEGG:00620 | Pyruvate metabolism | 5.4E−02 | 5.7E−02 |
| KEGG:00261 | Monobactam biosynthesis | 5.9E−02 | 5.8E−02 |
| KEGG:00250 | Alanine, aspartate, and glutamate metabolism | 5.9E−02 | 5.8E−02 |
| KEGG:00910 | Nitrogen metabolism | 5.9E−02 | 5.8E−02 |

[a]Enrichment analysis was performed in g:Profiler against the ascomycete *Aspergillus oryzae* (version e104_eg51_p15_3922dba [25 October 2021]) since the model organism *Laccaria bicolor* is not available in g: Profiler. Term indicates the KEGG pathways to which Enzyme Commission numbers are mapped, term name indicates the KEGG pathways, and *P* values are the FDR-adjusted *P* values for the ectomycorrhizal fungi or all fungi in the study.

in 75 and 74 differentially expressed beech gene models, respectively, with both treatments sharing 26 differentially expressed genes (DEGs) (Fig. 4A), indicating overlapping responses to ammonium and nitrate. Among these overlapping DEGs, a putative GS (AT5G35630.2) showed the highest upregulation, along with five putative cysteine-rich receptor-like protein kinase orthologs of *Arabidopsis thaliana* (*CRK8*; AT4G23160.1), outward rectifying potassium channel protein (*ATKCO1*; AT5G55630.2), HXXXD-type acyl transferase family protein (AT5G67150.1), hemoglobin 1 (*HB1*; AT2G16060.1), molybdate transporter 1 (*MOT1*; AT2G25680.1), and early nodulin-like protein 20 (*ENODL20*; AT2G27035.1) (Fig. 4B). Moreover, among the downregulated overlapping DEGs were a cinnamate-4-hydroxylase (*C4H*; AT2G30490.1), which plays a role in plant phenylpropanoid metabolism, growth, and development (61); eight orthologs coding for DNase 1-like superfamily proteins (AT1G43760.1); AP2/B3-like transcription factor family proteins (*VRN1*; AT3G18990.1), which are involved in the regulation of the vernalization pathway (62, 63); a subtilase family protein (AT5G45650.1); an ankyrin repeat family protein (AT3G54070.1); LRR and NB-ARC domain-containing disease resistance protein (*LRRAC1*; AT3G14460.1), known to play roles in the immune response against biotrophic fungi and hemibiotrophic bacteria (64); and an NB-ARC domain-containing disease resistance protein (AT4G27190.1) (Fig. 4B).

Among the unique responses to ammonium treatment were the upregulation of a further *GS* ortholog (AT5G35630.2) and the downregulation of a putative nitrate transporter gene (*NRT1.5*; AT1G32450.1) (Fig. 4B) known to load nitrate into the xylem and to be induced at high or low nitrate concentrations in *Arabidopsis thaliana* (65). Among the unique DEGs detected in response to nitrate treatment, and known to play roles in nitrate translocation and metabolism, were a putative high-affinity nitrate transporter (*NRT3.1*; AT5G50200.1), which was upregulated along with genes encoding a putative nitrite transmembrane transporter (*ATNITR2;1*; AT5G62720.1 [see reference 66]), nitrite reductase 1 (*NIR*; AT2G15620.1), molybdate transporter 1 (*MOT1*; AT2G25680.1), SLAC1 homolog 3 (*SLAH3*; AT5G24030.1), and chloride channel b (*CLC-B*; AT3G27170.1) (Fig. 4B). Furthermore, transcripts for the root-type ferredoxin:NADP(H) oxidoreductase gene (*RFNR1*; AT4G05390.1), which supplies electrons to ferredoxin-dependent enzymes (e.g., Fd-NiR and Fd-GOGAT) (67), and a ferredoxin 3 gene (*FD3*; AT2G27510.1), which enables electron transfer activity, were also upregulated, while a putative nitrate transporter

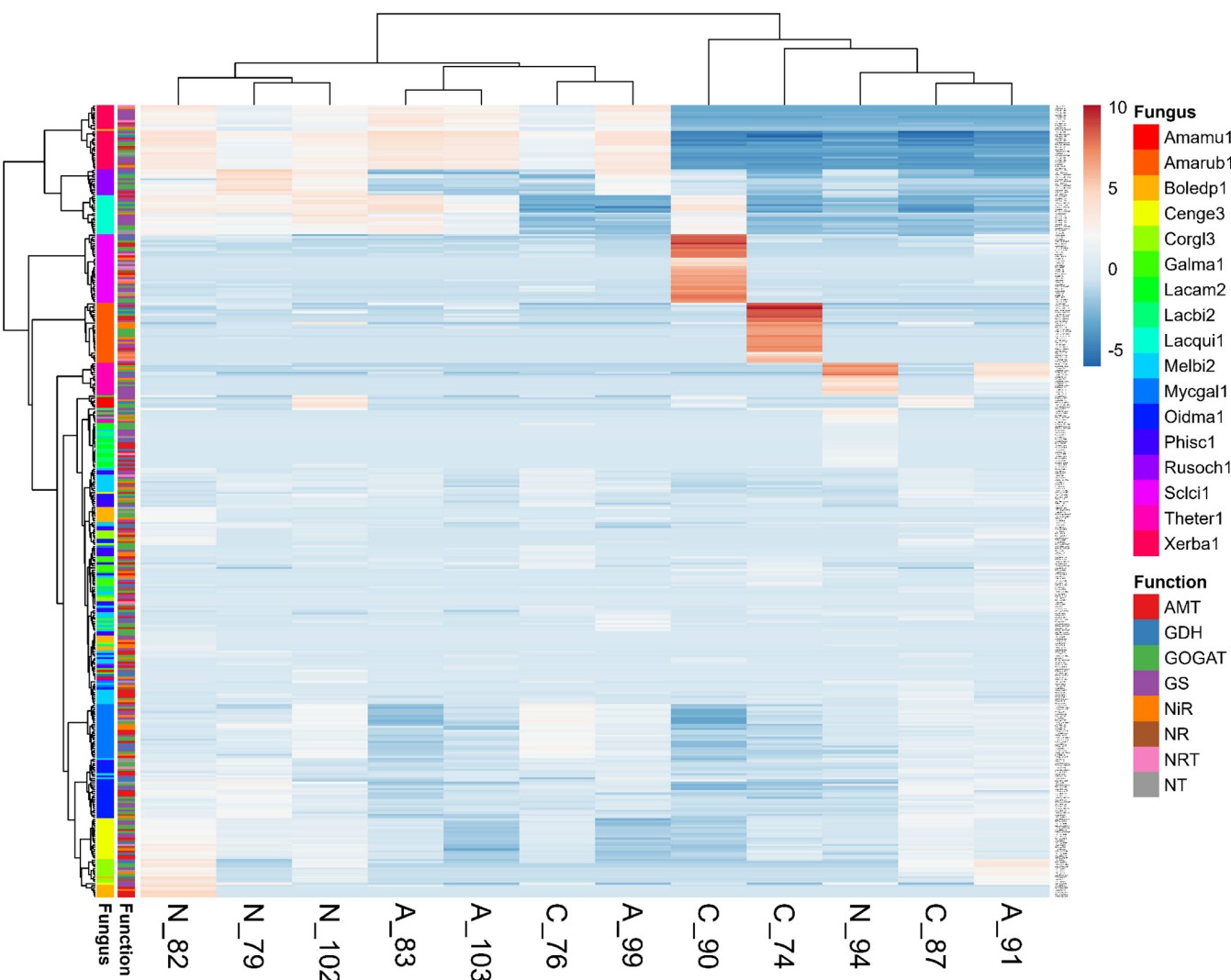

**FIG 3** Cluster analysis of N-related transporters and enzymes represented by transcript abundances for ectomycorrhizal, endophytic, and saprotrophic fungi colonizing beech roots. Samples are indicated with C for the control, A for ammonium treatment, and N for nitrate treatment. Abbreviations for fungi (from the top down) are *Amanita muscaria*, *Amanita rubescens*, *Boletus edulis*, *Cenococcum geophilum*, *Cortinarius glaucopus*, *Galerina marginata*, *Laccaria amethystina*, *Laccaria bicolor*, *Lactarius quietus*, *Meliniomyces bicolor*, *Mycena galopus*, *Oidiodendron maius*, *Phialocephala scopiformis*, *Russula ochroleuca*, *Scleroderma citrinum*, *Thelephora terrestris*, and *Xerocomus badius*. Original values of the transcript levels were ln(*x* + 1) transformed. Rows are centered; no scaling is applied to rows. Both rows and columns are clustered using Euclidean distance and Ward linkage. There are 369 rows and 12 columns. Fungal clusters (Fungus) and genes forming the cluster (Function) are shown at the left.

gene (*NRT1/PTR FAMILY 6.2*; AT2G26690.1) was downregulated (Fig. 4B). Other genes involved in N assimilation exhibited basal transcript levels, including those coding for the enzymes GOGAT and GDH, which were detected under nitrate, ammonium, and control conditions but not differentially regulated.

Classification of beech DEGs into MapMan bins revealed a significant overrepresentation of genes involved in "nitrogen metabolism" for both ammonium and nitrate treatments (Fig. 5). Significantly overrepresented metabolic processes for the nitrate treatment included "oxidative pentose phosphate pathway" (OPP), "protein," "redox," "secondary metabolism," "signaling," and "stress" (Fig. 5). For the ammonium treatment, significantly overrepresented functions included "DNA," "hormone metabolism," "secondary metabolism," "signaling," "stress," and "transport" (Fig. 5). Pathway enrichment analysis of Gene Ontology (GO) terms of beech DEGs in g:Profiler returned significant results for nitrate but not for ammonium treatment. DEGs from the nitrate treatment resulted in 38 significantly enriched GO terms involving nitrate-related molecular-level functions and 4 biological processes, including "nitrate transmembrane

**TABLE 3** Biomass and root and soil chemistry in control and $^{15}$N-ammonium- or $^{15}$N-nitrate-treated cosms[a]

| Variable | Mean value ± SD | | | F value | P value |
|---|---|---|---|---|---|
| | Control | Ammonium | Nitrate | | |
| Biomass of CR (g cosm$^{-1}$) | 3.44 ± 1.37 A | 2.86 ± 1.21 A | 3.40 ± 1.16 A | 0.5395 | 0.5905 |
| Biomass of FR (g cosm$^{-1}$) | 0.88 ± 0.47 A | 0.66 ± 0.41 A | 0.70 ± 0.28 A | 0.6865 | 0.5138 |
| Biomass of EMRTs (g cosm$^{-1}$) # | 0.22 ± 0.17 A | 0.16 ± 0.06 A | 0.15 ± 0.08 A | 0.6581 | 0.5277 |
| Soil dry mass (g cosm$^{-1}$) | 1,227 ± 202 A | 1,149 ± 310 A | 1,155 ± 466 A | 0.141 | 0.8693 |
| $^{15}$N enrichment (mg g$^{-1}$ CR) | NA | **0.11 ± 0.03 B** | **0.06 ± 0.02 A** | **9.8675** | **0.008512** |
| $^{15}$N enrichment (mg g$^{-1}$ FR) # | NA | 0.27 ± 0.11 A | 0.21 ± 0.04 A | 1.1993 | 0.295 |
| $^{15}$N enrichment (mg g$^{-1}$ EMRT) # | NA | 0.64 ± 0.45 A | 0.52 ± 0.11 A | 0.0285 | 0.8768 |
| $^{15}$N enrichment (mg g$^{-1}$ soil) # | NA | 0.0171 ± 0.0062 A | 0.0237 ± 0.017 A | 0.8583 | 0.3725 |
| $^{15}$N enrichment in roots (mg cosm$^{-1}$) | NA | 0.53 ± 0.25 A | 0.42 ± 0.22 A | 0.7395 | 0.4067 |
| $^{15}$N enrichment in soil (mg cosm$^{-1}$) | NA | 18.35 ± 4.63 A | 20.76 ± 6.38 A | 0.6558 | 0.4338 |
| N (mg g$^{-1}$ CR) | 9.16 ± 2.28 A | 10.48 ± 2.29 A | 9.19 ± 1.77 A | 0.9419 | 0.4065 |
| N (mg g$^{-1}$ FR) | **12.90 ± 1.72 A** | **14.68 ± 1.60 AB** | **15.13 ± 1.36 B** | **4.5612** | **0.02334** |
| N (mg g$^{-1}$ EMRT) | 16.07 ± 4.83 A | 17.44 ± 0.06 A | 18.69 ± 1.85 A | 0.4571 | 0.6507 |
| N (mg g$^{-1}$ soil) $ | 4.34 ± 3.06 A | 3.98 ± 2.93 A | 4.61 ± 3.85 A | 4E−04 | 0.9996 |
| C (mg g$^{-1}$ CR) | **450.95 ± 5.74 AB** | **456.03 ± 8.04 B** | **444.97 ± 7.16 A** | **4.4774** | **0.02472** |
| C (mg g$^{-1}$ FR) | 479.61 ± 14.80 A | 472.13 ± 21.32 A | 467.36 ± 13.35 A | 1.1062 | 0.3502 |
| C (mg g$^{-1}$ EMRT) | 435.74 ± 93.60 A | 462.10 ± 1.65 A | 465.03 ± 9.07 A | 0.202 | 0.8217 |
| C (mg g$^{-1}$ soil) # | 114.99 ± 88.14 A | 104.78 ± 82.78 A | 123.77 ± 113.81 A | 0.0699 | 0.9327 |
| C/N ratio in CR | 51.93 ± 12.41 A | 45.32 ± 10.40 A | 50.00 ± 9.54 A | 0.7282 | 0.4951 |
| C/N ratio in FR # | **37.70 ± 4.75 B** | **32.38 ± 2.43 A** | **31.05 ± 2.20 A** | **7.8149** | **0.003106** |
| C/N ratio in EMRTs $ | 28.12 ± 6.45 A | 26.54 ± 0.00 A | 25.02 ± 2.05 A | 0.3095 | 0.7434 |
| C/N ratio in soil | 25.75 ± 2.90 A | 25.78 ± 2.21 A | 25.42 ± 2.75 A | 0.0423 | 0.9586 |
| N-NH$_4^+$ (mg g$^{-1}$ FR) | 0.11 ± 0.03 A | 0.09 ± 0.03 A | 0.09 ± 0.03 A | 0.3329 | 0.7253 |
| N-NO$_3^-$ (mg g$^{-1}$ FR) | 1.96 ± 0.32 A | 2.59 ± 0.73 A | 1.87 ± 0.46 A | 2.2306 | 0.1634 |
| Glucose (mg g$^{-1}$ FR) | 16.99 ± 2.73 A | 16.32 ± 2.20 A | 16.36 ± 1.98 A | 0.1062 | 0.9003 |
| Fructose (mg g$^{-1}$ FR) | 9.06 ± 1.23 A | 8.51 ± 1.84 A | 7.71 ± 0.90 A | 0.9712 | 0.415 |
| Sucrose (mg g$^{-1}$ FR) | 0.61 ± 0.47 A | 0.51 ± 1.02 A | 0.82 ± 1.64 A | 0.0759 | 0.9275 |
| Starch (mg g$^{-1}$ FR) # | 21.20 ± 11.95 A | 18.40 ± 8.66 A | 16.03 ± 5.20 A | 0.2411 | 0.7907 |
| TNSC (mg g$^{-1}$ FR) # | 47.87 ± 14.25 A | 43.75 ± 12.51 A | 40.93 ± 6.33 A | 0.3484 | 0.7149 |

[a]Analyses were conducted 2 days after watering each cosm with 35 mg $^{15}$N. The mean soil pH was 3.6 ± 0.1, and the mean relative soil water content was 47.6% ± 28.9% ($n = 25$) across all studied cosms. Data are shown as means ± standard deviations for dry samples. For dry mass, there were 9 samples for the control, 8 samples for ammonium, and 8 samples for nitrate treatments. For $^{15}$N, C, and N, there were 9 samples for the control, 7 samples for ammonium, and 7 samples for nitrate treatments, except for root tips, where there were 5 samples for the control, 2 samples for ammonium, and 3 samples for nitrate treatments. For ammonium-N, nitrate-N, and nonstructural carbohydrates in fine roots, there were 4 samples per treatment. Significant differences among treatments (control, ammonium, and nitrate) at a P value of <0.05 (Tukey's HSD test) are shown in rows and marked in boldface type. Different letters denote detectable differences between conditions, and same letters denote no detectable difference between the treatment conditions, row-wise. Abbreviations: CR, coarse roots; FR, fine roots; EMRTs, ectomycorrhizal root tips; TNSC, total nonstructural carbohydrates; NA, not applicable because the mean $^{15}$N values of nonlabeled controls were subtracted from the values of the $^{15}$N-treated samples. Symbols indicate whether the data were log transformed (#) or inverse transformed ($).

transporter activity," "nitrite reductase activity," "response to nitrate," and "nitrate transport" (Table S3). Plant immune responses induced by nitrate were also evident via the enrichment of a putative isochorismate synthase gene (*ICS2*; AT1G18870) and a flavin-dependent monooxygenase 1 gene (*FMO1*; AT1G19250). *ICS2* is involved in the biosynthesis of vitamin K$_1$ (68) and potentially in salicylic acid biosynthesis (69, 70). *FMO1* is involved in the catalytic conversion of pipecolic acid to *N*-hydroxypipecolic acid (NHP), which plays a role in plant-acquired systemic resistance to infection by pathogens (71).

## DISCUSSION

**Ectomycorrhizal and root acquisition of nitrate or ammonium.** A central aim was to gain insights into gene regulation in naturally assembled ectomycorrhizas by targeting the transcriptomes of the EMF living in active symbiosis with beech roots in response to N provision. To challenge fungal metabolism, we applied N treatments that caused about 3- and 14-fold increases in the available NH$_4^+$-N (about 15.1 $\mu$g g$^{-1}$ [dry weight] of soil) and NO$_3^-$-N (about 2.3 $\mu$g g$^{-1}$ [dry weight] of soil), respectively. The magnitude of these variations was similar to the temporal fluctuations of NH$_4^+$-N and NO$_3^-$-N observed in the soil of beech stands, with 2-fold and 10-fold changes for NH$_4^+$-N and NO$_3^-$-N, respectively (72). Therefore, it was expected to elicit representative N

mSystems®

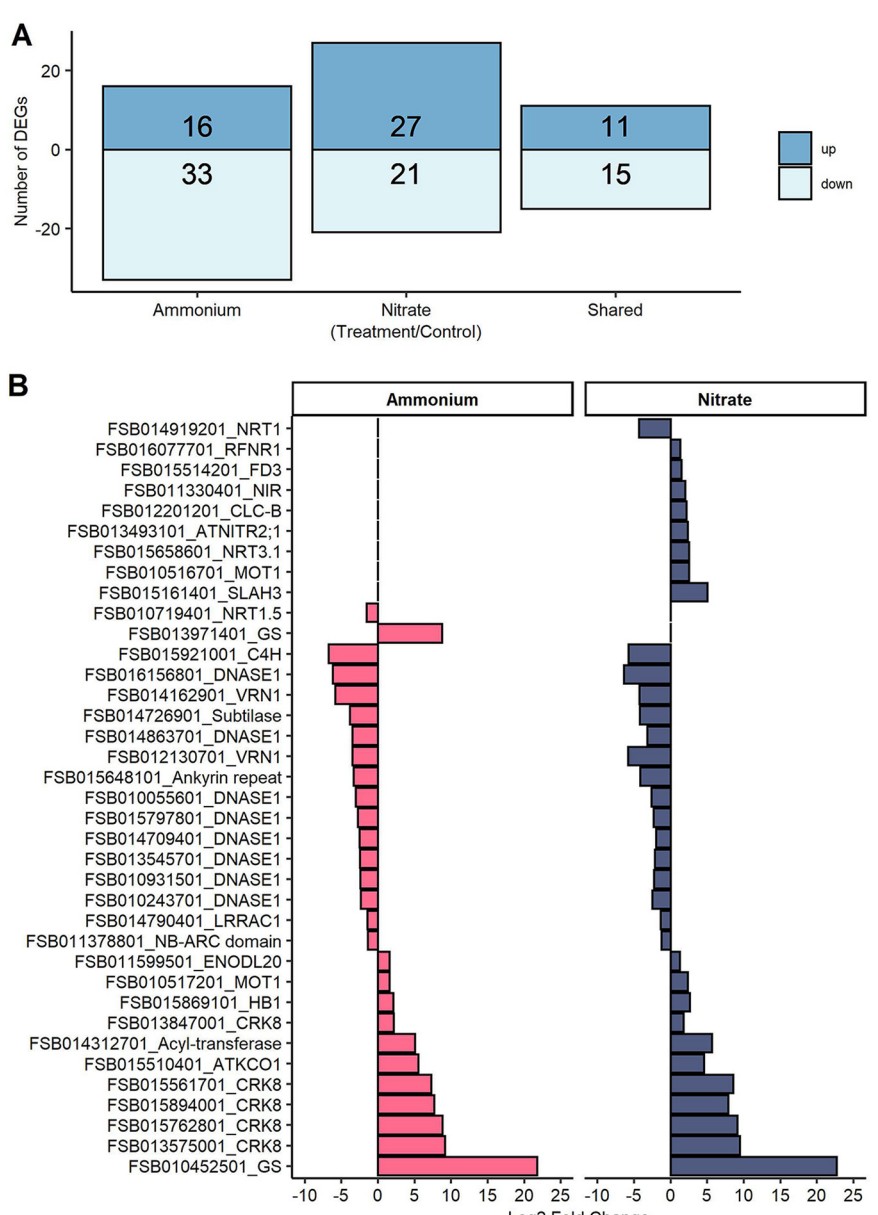

**FIG 4** Differentially expressed genes (DEGs) in response to ammonium or nitrate exposure. (A) Numbers of unique and shared DEGs (FDR-adjusted $P$ value of <0.05 and 2-fold change) in response to ammonium or nitrate treatment. (B) $Log_2$ fold changes of shared DEGs and DEGs related to N metabolisms in beech roots in response to increased ammonium or nitrate treatment relative to control conditions ($n = 4$ per treatment). The complete information, gene model identifiers, and names are provided in Data Set S5 at Dryad (132).

responses in the naturally assembled EMF communities. The EMF assemblages in our study showed the typical patterns known for temperate beech forests, with high diversity (21, 23, 24), a dominance of certain species (73, 74) (e.g., the genera *Amanita*, *Xerocomus*, and *Scleroderma* in this study), and nonuniform occurrence in the tree roots. However, the 2-day N treatments were not expected to affect the fungal community structure because the colonization and establishment of new ectomycorrhizas take weeks or months rather than days (75, 76), and shifts in fungal communities toward more nitrophilic fungi occur as a consequence of long-term exposure to high N loads (77–81).

The EMF community in this study was composed of taxa characteristic of acidic, sandy, nutrient-poor soils, including genera in the orders Agaricales, Boletales, Russulales, Helotiales,

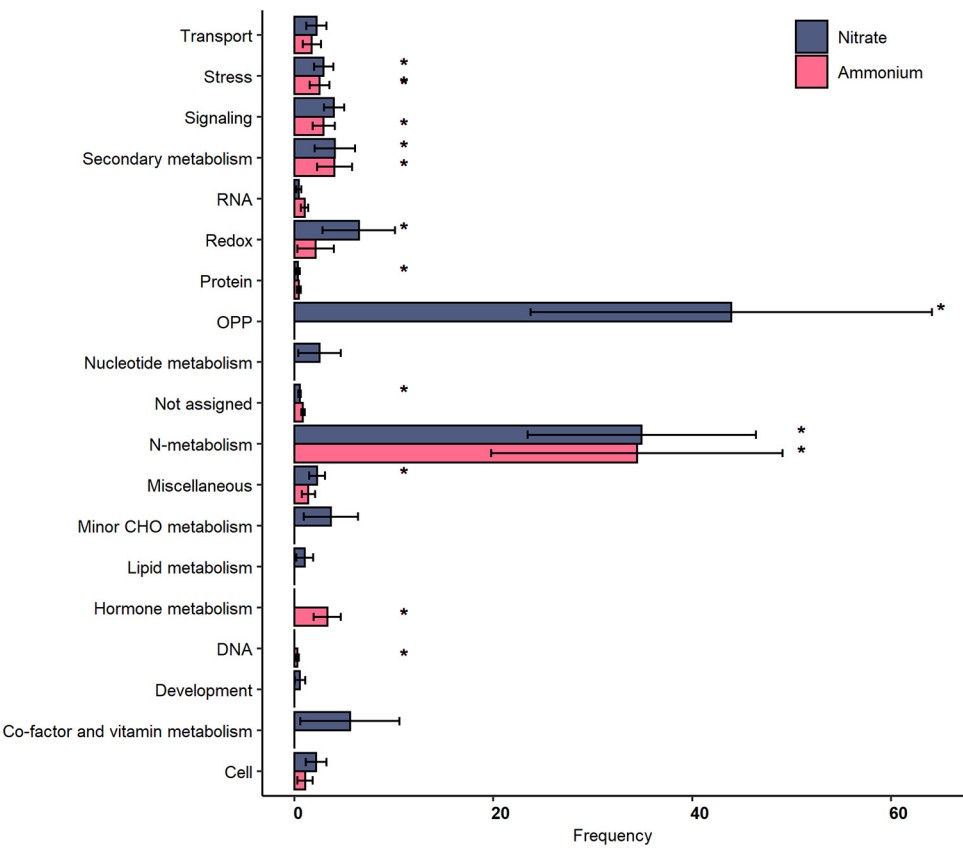

**FIG 5** Classification of beech root DEGs in response to nitrate or ammonium treatment. Genes were classified according to MapMan bins using the Classification SuperViewer in BAR (http://bar.utoronto.ca/ntools/cgi-bin/ntools _classification_superviewer.cgi). Bins that were statistically significantly enriched are marked with an asterisk.

Mytilinidales, and Thelephorales (see Data Set S1 at Dyad). These fungi vary in their foraging strategies, being equipped with different types of hyphae for scavenging N. *Cenococcum geophilum*, which is a widespread fungus and known for its tolerance to drought (82), produces short- and medium-distance hyphae, while the hyphae of *Amanita* (medium-distance smooth or long distance), *Cortinarius* (medium-distance fringe), *Laccaria* and *Thelephora* (medium-distance smooth), *Lactarius* (contact, short, and medium distance), *Russula* (contact), *Scleroderma* (long distance), and *Xerocomus* (long distance) possess diverse hyphal lengths (78, 83). EMF that produce hydrophilic hyphae of the contact, short-distance, and medium-distance smooth exploration types were reported to respond positively or to display a mixed response to mineral N enrichment, whereas EMF with medium-distance fringe hydrophobic hyphae are the most sensitive, and those with long-distance hydrophobic hyphae vary in their responses to mineral N (78).

In this experiment, the availability of either nitrate or ammonium was suddenly increased to simulate fluctuations that roots and microbes must handle during N nutrition. Generally, the negative charge of nitrate ions in the soil solution makes it more readily available for uptake by the roots, whereas the positively charged ammonium ions tend to be fixed by soil colloids; however, nitrate is more prone to leaching in sandy soils, while ammonium retention by organic matter and clay minerals is generally higher (5, 13, 14, 84). The lower energy costs needed for ammonium metabolism make its utilization more advantageous than nitrate. This was previously observed in EMF (30–35), and in agreement, we found higher translocation of $^{15}N$ from $NH_4^+$ than from $NO_3^-$ to the coarse roots. The enrichment of the newly applied $^{15}N$ in the EMRTs was strong but did not differ between the N forms applied. We cannot exclude ammonification by soil microbes, potentially converting $NO_3^-$ to $NH_4^+$ in the soil before its

uptake by the EMF, thus contributing to similar $^{15}$N accumulation patterns in the ecto-mycorrhizas after nitrate or ammonium application. Microbial turnover rates are estimated to be about 24 h for ammonium and a few days for nitrate (85). However, the significant transcriptional regulation of nitrate marker genes in beech roots under nitrate exposure supports that $NO_3^-$ was taken up by the root system. In fine root cells, $NO_3^-$ was considerably more abundant than $NH_4^+$, as observed in beech trees under field conditions (72, 86), and unaffected by mineral N addition. Our results demonstrate that the newly acquired $^{15}$N was metabolized because the root N concentrations, but not the levels of ammonium or nitrate, increased.

**N assimilation uncovers fungal taxon-specific but not N-induced transcription patterns in root-associated fungal communities.** Despite the compelling support for N uptake and assimilation in roots, the EMF metatranscriptome did not show any significant changes related to N metabolism, and similar results were observed when the full fungal metatranscriptome list was considered in the analysis. Initially, we hypothesized that if the root-associated fungi (RAF) and the beech root cells responded like a synchronized "superorganism," both fungi and roots would show similar patterns of transcriptional regulation. However, this hypothesis is rejected because N-responsive DEGs were found in beech but not in the EMF metatranscriptome or the full fungal metatranscriptome, except for KOG4381 and KOG4431, which were induced by nitrate. Closer inspection revealed that KOG4381 occurred in only two ectomycorrhizal fungi (*Thelephora terrestris* and *Russula ochroleuca*), thus not reflecting a community response and rather suggesting that in the symbiotic system, the host and EMF partners respond as individual autonomous units. KOG4431 was present in nine EMF (*Laccaria amethystina*, *Meliniomyces bicolor*, *Russula ochroleuca*, *Scleroderma citrinum*, *Thelephora terrestris*, *Xerocomus badius*, *Amanita rubescens*, *Boletus edulis*, and *Cenococcum geophilum*), one saprotroph (*Mycena galopus*), and one ericoid mycorrhizal fungus (*Oidiodendron maius*). Further analyses are needed to clarify the roles of these two KOGs in nitrate signaling. Although differentially expressed KOGs were rare in both the EMF metatranscriptome and the full fungal metatranscriptome list, putative nitrate/nitrite transporters, ammonium transporters and enzymes (NR, NiR, GS, GOGAT, and GDH) were transcribed (Fig. 6), representing all necessary steps for mineral N uptake and assimilation into amino acids (AA). In controlled laboratory studies, many of these transporters and enzymes have been characterized in EMF and were regulated by N form and availability, for instance, high-affinity nitrate/nitrite transporters (NRT2), nitrate reductase (NR), and nitrite reductase (NiR1) in *Hebeloma cylindrosporum* (87, 88); NRT2, NR1, and NiR1 in *Tuber borchii* (89, 90); NRT, NR, and NiR in *Laccaria bicolor* (44, 91); high- and low-affinity ammonium transporters (AMT1, AMT2, and AMT3) in *Hebeloma cylindrosporum* (50, 92); AMT2 in *Amanita muscaria* (93); and AMT1, AMT2, and AMT3 in *Laccaria bicolor* (44). Although we did not find N-induced regulation of specific genes, KEGG pathway enrichment analysis shows that functions related to N assimilation and carbon metabolism were represented across all studied fungi. We suggest that at the whole EMF community level, primary metabolism is genetically equipped for handling fluctuating environmental N availability and host-derived C supply.

The observed stability of the fungal metatranscriptomes was unexpected because stable-isotope labeling and electrophysiological studies showed a distinct responsiveness of different fungal taxa to environmental changes in naturally assembled communities (27, 28, 94), and controlled studies (described above and in the introduction) showed significant regulation of N-related genes. Our study does not exclude that there were N-induced responses in distinct fungi, but weak effects might have been masked by the heterogeneous occurrence of EMF in roots of individual plants. Presumed species-specific responses to N fertilization were probably also overridden by interspecific differences. This can be inferred from the observation that arrays of N-related genes clustered quite strictly according to fungal species but not according to the genes with similar functions. Our identification of expression patterns for the fungi under study is an important, novel result underpinning trait stability within naturally assembled EMF in beech roots.

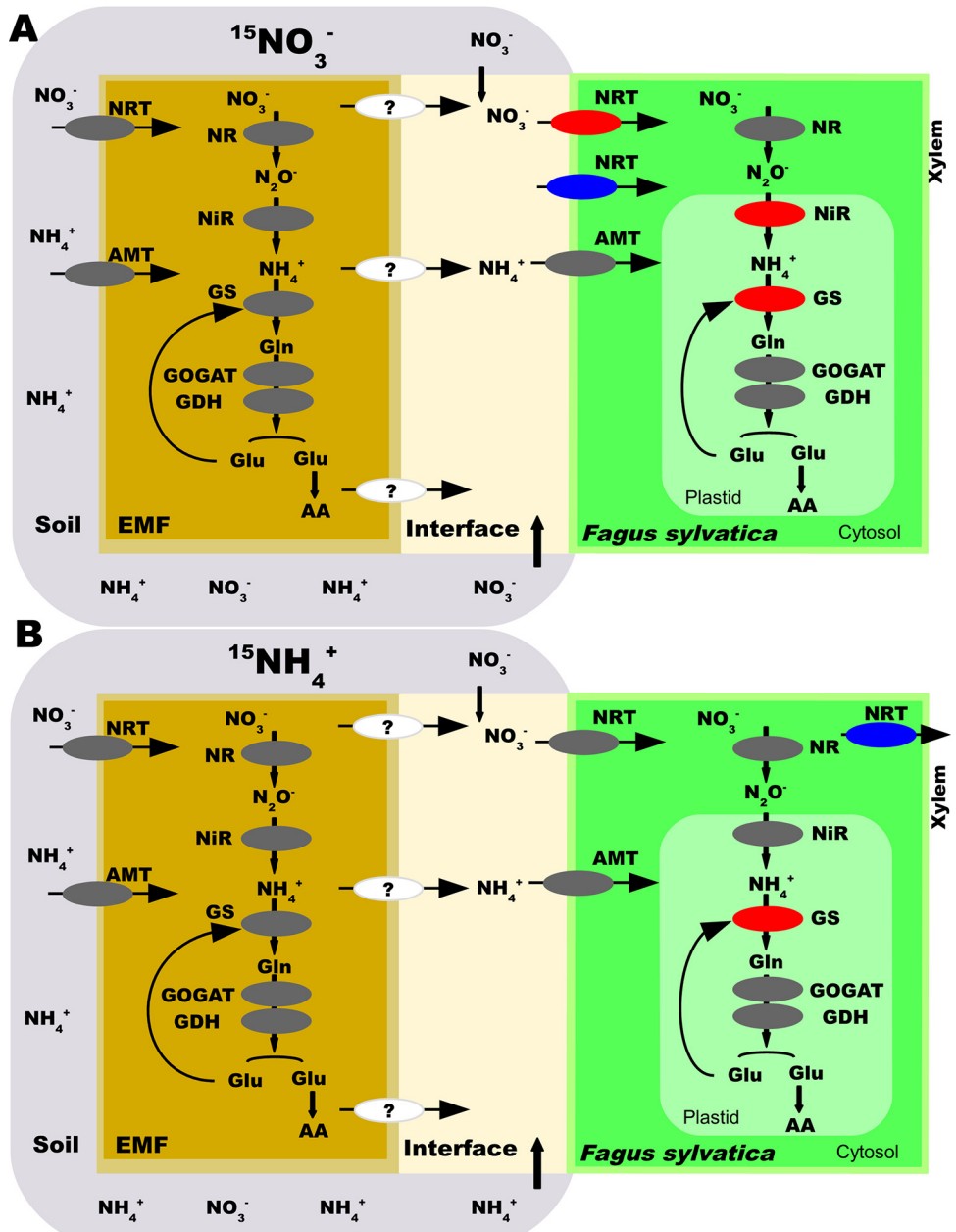

**FIG 6** Scheme of the pathway for N uptake and assimilation in EMF and *Fagus sylvatica* based on transcription profiles. The regulation of ectomycorrhizal fungus and host transcripts encoding transporters and enzymes involved in N uptake and assimilation detected in the nitrate treatment (A) and in the ammonium treatment (B) was determined. NRT, nitrate/nitrite transporter; NR, nitrate reductase; NiR, nitrite reductase; AMT, ammonium transporter; GS, glutamine synthetase; GOGAT, glutamate synthase; GDH, glutamate dehydrogenase; AA, amino acids. Gray, detected but not regulated; red, upregulated; blue, downregulated; white, not detected.

**Ammonium and nitrate induce specific assimilation patterns in beech roots.**
Our initial hypothesis was that EMF shield the plant cells against major fluctuations in N availabilities, and therefore, we expected no or moderate changes in the beech root transcriptome after N fertilization. This hypothesis is rejected since both ammonium and nitrate treatments caused drastic changes in the beech root transcriptome. The strategy of European beech for dealing with high loads of inorganic N availability was transcriptional upregulation of genes involved in N uptake and assimilation, as observed in *Arabidopsis* (51), a nonmycorrhizal species. The transcription patterns in response to nitrate and ammonium were clearly distinguishable, in agreement with other studies that documented nitrate- and ammonium-specific effects on gene

regulation, signaling, and lateral root growth (95–100). Notably, transcripts belonging to putative NRTs and to enzymes (NR, NiR, and GS) were significantly upregulated in the nitrate treatment encompassing the suite of reactions required for $NO_3^-$ reduction and incorporation into amino acids (Fig. 6). In addition, the upregulation of root ferredoxin and molybdate transporters pointed to an enhanced need for reducing power and the biosynthesis of NR, which requires molybdate in its active center (101). The significant activation of defenses against biotrophic fungi by nitrate was also remarkable. Similar results were shown in leaves of nonmycorrhizal nitrate-fed trees (102). In the ammonium treatment, a significant increase in the transcript levels of two GS enzyme isoforms was detected, while NRT1.5, potentially loading nitrate into the xylem, was downregulated. Remarkably, nitrate and ammonium treatments showed a common pattern, with strong upregulation of *GS* and *CRK*-like genes (Fig. 4B). CRK receptor kinases are involved in stress and plant pathogen responses and cell death (103, 104). The *Arabidopsis* ortholog CRK8 is regulated in senescing leaves (105), and while a function in N metabolism appears likely, controlled experiments are needed. Overall, these results were in line with the expectation that nitrate-specific, ammonium-specific, and overlapping responses would be found. We demonstrate for the first time that excessive N in EMRTs is actively metabolized by the plant. It remains unknown if $NO_3^-$ and $NH_4^+$ were taken up by EMF and transferred to the plant for further assimilation or if excessive N circumvented the fungal barrier, entering the plant directly (Fig. 6).

In conclusion, effects of high levels of ammonium or nitrate were not evident in the EMF metatranscriptome or the full fungal metatranscriptome, whereas the host tree responded to ammonium and nitrate by upregulating genes involved in the assimilation of the surplus inorganic N into organic forms. Although it is unknown whether the applied $^{15}N$ sources underwent conversions due to microbial activities, the response of European beech indicated that a significant proportion of ammonium and nitrate was taken up in the originally added form. The fungal transcriptomes suggested species-specific metabolic responses to N, implying significant trait stability for N turnover and suggesting that EMF in temperate beech forests are resistant to short-term fluctuations in environmental mineral N pools. However, further work is required to investigate to what extent this tolerance capacity can be sustained and its ecological relevance under chronic N exposure.

## MATERIALS AND METHODS

**Tree collection, maintenance, and experimental setup.** European beech (*Fagus sylvatica* L.) saplings were collected on 7 March 2018 in a 122-year-old beech forest (53°07′27.7″N, 10°50′55.7″E; 101 m above sea level [Göhrde, Lower Saxony, Germany]). The soil type is podzolic brown earth with parent material consisting of fluvioglacial sands (106). In 2017, the mean annual temperature was 9.9°C, and the total annual precipitation was 768 mm, whereas on the day of tree collection, the mean air temperature was 4.6°C, and the precipitation was 0.66 mm (https://www.dwd.de). The beech saplings (*n* = 34) were excavated using polyvinyl chloride cylinders (diameter of 0.125 m and depth of 0.2 m), which were placed around a young tree, hammered into the ground to a depth of 0.2 m, and then carefully lifted to keep the root system in the intact forest soil. These experimental systems are referred to as cosms. The cosms were transported to the Forest Botanical Garden, University of Göttingen (51°33′27.1″N, 9°57′30.2″E), where they were maintained outdoors under a transparent roof and exposed to natural climatic conditions except for rain (see Table S4 in the supplemental material). A green shading net was placed over the roof to protect the trees from direct sun, similar to shading in the forest. Thereby, on average, full sunlight was reduced on sunny days from 1,125 $\mu$mol m$^2$ s$^{-1}$ to 611 $\mu$mol m$^{-2}$ s$^{-1}$ PAR (photosynthetically active radiation) and on cloudy days from 284 $\mu$mol m$^{-2}$ s$^{-1}$ to 154 $\mu$mol m$^{-2}$ s$^{-1}$ (quantum/radiometer/photometer model 185B; Li-Cor Inc., Lincoln, NE, USA). The cosms were regularly watered with demineralized water. Control of the water quality (Seal AutoAnalyzer 3 HR flow analyzer; Seal Analytical GmbH, Norderstedt, Germany) revealed 0.2 mg $NH_4^+$ liter$^{-1}$ and no detectable $NO_3^-$ in the irrigation water. The cosms were randomly relocated every other day to avoid confounding positional effects. The trees were grown under these conditions until July 2018. By this time, the trees had a mean height of 0.401 ± 0.08 m and a root collar diameter of 6.11 ± 0.95 mm. The trees were about 8 (±2) years old based on the number of growth scars along the stem (107). Before the $^{15}N$ treatments, ammonium and nitrate were measured in the soil (see Text S1 in the supplemental material for details). The cosms contained 15.1 ± 11.3 $\mu$g $NH_4^+$-N g$^{-1}$ (dry weight) of soil and 2.3 ± 1.3 $\mu$g $NO_3^-$-N g$^{-1}$ (dry weight) of soil (*n* = 3) (means ± standard deviations [SD]), equivalent to approximately 9.1 mg $NH_4^+$-N and 1.8 mg $NO_3^-$-N cosm$^{-1}$.

**Application of $^{15}N$-labeled ammonium and nitrate.** Before labeling, the even distribution of the irrigation solution in the soil was tested on separate cosms using blue dye (GEKO Lebensmittelfarbe,

Wolfram Medenbach, Gotha, Germany) in water. The experimental cosms were assigned the following treatments: control (no nitrogen application), $^{15}NH_4^+$ application, or $^{15}NO_3^-$ application. The cosms were surface irrigated at 7 a.m. with 60 ml of either 19.85 mM $^{15}NH_4Cl$ (99% $^{15}N$; Cambridge Isotope Laboratories Inc., MA, USA) (pH 5.47) or a 19.98 mM $^{15}KNO_3$ (99% $^{15}N$; Cambridge Isotope Laboratories) (pH 6.23) solution prepared in autoclaved deionized water. Controls were irrigated with 60 ml autoclaved demineralized water (pH 6.07). Each of these treatments was repeated the next day, resulting in a total application of 35.96 mg $^{15}N$ in the nitrate-treated cosms or 35.74 mg $^{15}N$ in the ammonium-treated cosms, corresponding to mean additions of approximately 30 $\mu$g $^{15}N$ g$^{-1}$ dry soil. Treatments were conducted in two batches: batch 1 included $^{15}N$ application on 17 July 2018 and harvest on 19 July 2018 ($n = 9$ cosms), and batch 2 included $^{15}N$ application on 31 July 31 2018 and harvest on 2 August 2018 ($n = 16$ cosms).

**Cosm harvest.** The cosms were harvested in the morning 48 h after the initial $^{15}N$ application in alternating order according to treatment. The tree-soil compartment was pushed out of the cylinder, collecting all parts. Roots were briefly rinsed with tap water and then with deionized water and gently surface dried with paper towels. The root tips were clipped off, shock-frozen in liquid nitrogen, and stored at −80°C. Aliquots of fine roots were shock-frozen in liquid nitrogen and stored at −80°C and −20°C, and soil aliquots were stored at −20°C. During the harvests, the fresh masses of all fractions (leaves, stem, coarse roots, fine roots, root tips, and soil) were recorded, and aliquots were taken for dry-to-fresh-mass determination after drying at 40°C (leaves, stems, and soil) or after freeze-drying (coarse roots, fine roots, and root tips). Biomass and soil mass in the cosms were calculated as total dry mass (g) = (total fresh weight × aliquot dry weight)/(aliquot fresh weight).

**Soil and root chemistry.** Soil pH was measured with a 538 pH meter (WTW, Weilheim, Germany) using a ratio of dry sieved soil to water of 1:2.5 according to the forestry analytics manual (108). The water content in the soil was calculated as relative soil water content (%) = (fresh soil weight − dry soil weight)/(dry soil weight) × 100.

For $^{15}N$ analyses, freeze-dried aliquots of soil, root tips, and fine and coarse root samples from both experimental batches were milled using a ball mill (type MM400; Retsch GmbH, Haan, Germany) in stainless steel grinding jars at a frequency of 30 Hz s$^{-1}$ in 20-s intervals to avoid heating the sample. The powder (control samples, 1.5 to 2 mg plant tissues and 5 mg soil; labeled samples, 1.5 to 3 mg plant tissue and 5 to 13 mg soil) was weighed into tin capsules (IVA Analysentechnik GmbH & Co. KG, Meerbusch, Germany) and measured at Kompetenzzentrum Stabile Isotope, Göttingen, Germany. The $^{15}N$ samples were measured in an isotope mass spectrometer (Delta V Advantage; Thermo Electron, Bremen, Germany) and an elemental analyzer (Flash 2000; Thermo Fisher Scientific, Cambridge, UK), and the nonlabeled control samples were measured using a mass spectrometer (Delta plus; Finnigan MAT, Bremen, Germany) and an elemental analyzer (NA1110; CE-Instruments, Rodano, Milan, Italy). Acetanilide (10.36% N and 71.09% C; Merck KGaA, Darmstadt, Germany) was used as the standard. Enrichments of $^{15}N$ in the ectomycorrhizal root tips (EMRTs), fine roots, coarse roots, and soil were calculated as $^{15}N$ enrichment (mg g$^{-1}$ [dry weight]) = APE/100 × N concentration of the sample (g g$^{-1}$ [dry weight]) × 1,000, where APE (atoms percent excess) = atom% $^{15}N$-labeled sample − atom% non-$^{15}N$-labeled sample and atom% $^{15}N$ = $(^{15}N)/(^{14}N + ^{15}N) × 100$.

For the determination of $NH_4^+$, $NO_3^-$, and nonstructural carbohydrates, 12 samples ($n = 4$ per treatment) of frozen fine roots (−80°C) were milled (MM400; Retsch GmbH) under liquid nitrogen to avoid thawing. For mineral N determination, the frozen powder (approximately 55 mg per test) was extracted as described previously (109), with slight modifications, and measured spectrophotometrically with the Spectroquant nitrate (catalog number 1.09713.0002) and ammonium (catalog number 1.14752.0002) test kits (Merk KGaA, Darmstadt, Germany). Glucose, fructose, sucrose, and starch were extracted from approximately 75 mg root powder and measured enzymatically as described previously (110). Details of all procedures are reported in Text S1.

**DNA extraction, Illumina sequencing, bioinformatic processing, and data analyses of fungi.** EMRTs previously stored at −80°C were homogenized in liquid nitrogen using a sterilized mortar and pestle. Each powdered, frozen sample was split into two parts: one for DNA extraction and Illumina sequencing of the fungal ITS2 marker gene and the other for RNA extraction and mRNA sequencing. DNA was extracted from approximately 200 mg of EMRT powder using the innuPREP plant DNA kit (Analytik Jena AG, Jena, Germany). Extraction, purification, processing, and sequencing are described in detail in Text S1. Briefly, the fungal nuclear ribosomal DNA internal transcribed spacer 2 (ITS2) region was amplified by PCR using the primer pair ITS3_KYO2 (111) and ITS4 (112), both containing specific Illumina overhang adapters (in italics; primers are underlined): 5'-*TCGTCGGCAGCGTCAGATGTGTATAAGAGACAG*GATGAAGAACGYAGYRAA-3' (forward [Miseq_ITS3_KYO2]) and 5'-*GTCTCGTGGGCTCGGAGATGTGTATAAGAGACAG*TCCTCCGCTTATTGATATGC-3' (reverse [Miseq_ITS4]).

After the PCR, the amplicons were purified and sequenced on a MiSeq flow cell using reagent kit v3 and 2-by-300 paired-end reads (Illumina Inc., San Diego, CA, USA), according to the manufacturer's instructions, at the Göttingen Genomics Laboratory (Institute of Microbiology and Genetics, Georg August University Göttingen, Göttingen, Germany). The raw sequences were quality filtered, merged, size filtered, denoised, and chimera checked. These high-quality sequences were clustered at 97% sequence identity into operational taxonomic units (OTUs), and abundance tables were generated. Taxonomic assignment of OTUs was first carried out against the UNITE database v8.2 (04.02.2020) (113), and BLAST analysis of all unidentified OTUs (114) was then performed against the nt (nucleotide) database (17 January 2020) to identify nonfungal OTUs. Two nonfungal OTUs were discarded from the taxonomic table. Ectomycorrhizal fungi and other ecological lifestyles of the fungal genera were identified using the FUNGuild annotation tool (115). Initially, 23 samples and 2 controls (positive and negative)

were sequenced; however, we did not find evidence of reagent contamination in the negative control, and only the 12 samples for which RNA sequencing was done were included for further analyses. The sequencing depth per sample was controlled and rarefaction analysis was conducted using the ampvis2 package (116). The samples were normalized by rarefying to the sample with the lowest sequencing depth (i.e., 20,051 reads). An overview of the sequence processing results is provided in Table S5, and the rarefied abundance table with taxonomic and ecological lifestyle assignments of the fungal OTUs is provided in Data Set S1 at Dryad (132).

**RNA extraction, library preparation, sequencing, and bioinformatic processing of the fungal metatranscriptome and beech transcriptome.** Total RNA was isolated from 25 of the frozen powdered beech root tip samples using an extraction method based on hexadecyltrimethylammonium bromide (117). Details are reported in Text S1. RNA integrity check, library preparation, and sequencing were conducted at Chronix Biomedical GmbH (Göttingen, Germany). Twelve samples with RNA integrity numbers ranging from 6.7 to 7.9 were selected for poly(A) selection and mRNA library preparation (Table S6). These samples also have the corresponding ITS2 metabarcoding sequencing data and are all from the same experimental batch. Libraries were constructed with the NEBNext RNA Ultra II library prep kit for Illumina (New England BioLabs, Ipswich, MA, USA) from 1 $\mu$g of purified RNA according to the manufacturer's instructions. Single-end reads with a length of 75 bp were sequenced on a NextSeq 500 sequencing system instrument (Illumina, San Diego, CA, USA) with a sequencing depth of 100 million reads per sample. Since there was no amplification in the negative control of the final library PCR, the negative control was not sequenced.

Processing (trimming, quality filtering, and adapter removal) of the raw sequence data (ca. 110 million reads per sample) resulted in approximately 109 million reads per sample (Table S6). The reads were mapped against the reference transcriptomes of *Fagus sylvatica* and 17 fungal species belonging to the same genera as those detected in the samples by ITS2 barcoding (Table 1). Reference beech sequences and annotations were downloaded from beechgenome.net (57), and reference fungal sequences and annotations were downloaded from the JGI MycoCosm database (56). The resulting 18 fasta files were concatenated to a single file, which was used to create an index file with bowtie2-build (118). The reads were mapped against this index file using bowtie2, resulting in one count table containing the reads for beech and fungi. On average, 61% of the reads could be mapped (45% to beech and 16% to fungi) (Table S6). The raw count table was split into a beech transcriptome count table and a fungal transcriptome count table. Normalization of the raw count tables and differential expression analyses relative to the control were conducted using the DESeq2 package (119), implemented in R (120). Differential expression analysis of the fungi was performed at the metatranscriptome level (i.e., the fungal raw count tables were aggregated by their Eukaryotic Orthologous Groups of protein identifiers [KOGs] [https://img.jgi.doe.gov/]), dropping taxon-specific information for the gene models. This approach was taken to deal with the patchy nature of fungal occurrence within replicates while improving read coverage across treatments for comparisons. Two fungal metatranscriptomes were considered: the full fungal metatranscriptome list (17 fungi) and the ectomycorrhizal fungus-specific metatranscriptome (13 fungi). Gene models (for the European beech transcriptome) or KOGs (for the two fungal metatranscriptomic data sets) with a Benjamini-Hochberg false discovery rate (FDR)-adjusted $P$ value of <0.05 (121) and at least a 2-fold change were considered significantly differentially expressed gene models (DEGs) or significant KOGs. The Enzyme Commission numbers assigned to the ectomycorrhizal fungal metatranscriptome were mapped to the KEGG metabolic pathways against the ectomycorrhizal fungal model *Laccaria bicolor* in KEGG Mapper (122). Functional enrichment analysis of all fungal expressed genes was carried out in g:Profiler (123) against KEGG metabolic pathways with *Aspergillus oryzae* as the reference since the model ectomycorrhizal fungus *Laccaria bicolor* was not available. Since a main interest in our experiment was to obtain information on fungal N uptake and metabolism, we manually searched the complete fungal metatranscriptomic database (see Data Set S2 at Dryad [132]) for N-related transporters and enzymes using the keywords "nitrate transporter," "nitrate reductase," "nitrite transporter," "nitrite reductase," "ammonium transporter," "glutamine synthetase," "glutamate synthase," and "glutamate dehydrogenase." These terms were searched in the definition lines accompanying the annotations of each of the fungal transcripts: "kogdefline" (definition of the KOG identifiers), "ECnumDef" (definition of the EC number), "iprDesc" (description of the InterPro identifiers), and "goName" (description of the Gene Ontology term). Cluster analyses were done in Clustvis (124). Gene Ontology term enrichment analysis of beech DEGs was also performed in g:Profiler (123). In addition, overrepresentation analysis of biological pathways based on the MapMan bin classification (Ath_AGI_LOCUS_TAIR10_Aug2012) of beech DEGs was performed using the Classification SuperViewer tool (125) from the Bio-Analytic Resource for Plant Biology (http://bar.utoronto.ca/).

**Statistical analyses.** The fungal community data were Hellinger transformed and fitted to a nonmetric multidimensional scaling (NMDS) ordination based on Bray-Curtis dissimilarity using the vegan package version 2.5-6 (126) and the ggplot2 function (127) in R software (128). Permutational analysis of variance (adonis 2) was used to test if the treatments resulted in significant effects on the fungal community or transcript composition. Quasi-Poisson regression models were used for overdispersed count data (e.g., species richness), and general linear models were applied to normally distributed data, followed by Tukey's honestly significant difference (HSD) *post hoc* test with the multcomp package (129). For biomass and root and soil chemistry (Table 3), when necessary, the data were log or inverse transformed to meet a normal distribution. If not indicated otherwise, data are shown as means ($\pm$SD). Linear regression analysis was conducted in R (128). For cluster analysis of N-related transporters and enzymes for all the fungi in the metatranscriptomic data set (Fig. 3), the original transcript values were ln($x$ + 1) transformed. Details on data transformation are indicated in the figure legends or table footnotes. One

cosm from ammonium and one from nitrate treatment were excluded from the $^{15}$N analyses since the measured $^{15}$N values in soil were higher than the concentration of added $^{15}$N.

**Data availability.** Raw sequences from fungal ITS2 gene metabarcoding-Illumina sequencing are available in the Sequence Read Archive of the National Center for Biotechnology Information under BioProject accession number PRJNA736215 (130). Raw read data from RNA-seq are also available at the ArrayExpress database under accession number E-MTAB-8931 (131). Additional supporting data (Data Sets S1 to S6) are accessible in Dryad (132).

## SUPPLEMENTAL MATERIAL

Supplemental material is available online only.

**TEXT S1**, DOCX file, 0.1 MB.
**FIG S1**, TIF file, 0.1 MB.
**FIG S2**, PDF file, 0.8 MB.
**TABLE S1**, DOCX file, 0.02 MB.
**TABLE S2**, DOCX file, 0.03 MB.
**TABLE S3**, DOCX file, 0.03 MB.
**TABLE S4**, DOCX file, 0.02 MB.
**TABLE S5**, DOCX file, 0.02 MB.
**TABLE S6**, DOCX file, 0.03 MB.

## ACKNOWLEDGMENTS

This research was funded by the German Research Foundation (DFG) through Research Training Group 2300 Enrichment of European Beech Forests with Conifers (contract number 316045089, project SP4). C.A.R.P. was financially supported by RTG2300.

We thank the Göhrde State Forest management office for authorizing tree collection in the forest. C.A.R.P. is grateful to Serena Müller for assistance in coordinating field work; Michael Reichel, Ronny Thoms, and Jonas Glatthorn for help collecting the trees in the forest; and Gaby Lehmann for help measuring ammonium/nitrate in soil.

C.A.R.P., Conceptualization, Data Curation, Formal Analysis, Investigation, Methodology, Project Administration, Visualization, Writing – Original Draft, Writing – Review & Editing; D.J., D.S., and R.D., Data Curation, Writing – Review & Editing; A.P., Conceptualization, Formal Analysis, Methodology, Supervision, Writing – Review & Editing.

We declare no competing interests.

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
