## [Reviewer comments · mSystems]

Transcriptional Landscape of Ectomycorrhizal Fungi and Their Host Provide Insight into N Uptake from Forest Soil

Carmen Rivera Pérez, Dennis Janz, Dominik Schneider, Rolf Daniel, and Andrea Polle

Corresponding Author(s): Carmen Rivera Pérez, Georg-August-University Göttingen

Review Timeline:

Submission Date:	July 24, 2021
Editorial Decision:	October 13, 2021
Revision Received:	November 7, 2021
Accepted:	November 29, 2021

Editor: Emily Cope

Reviewer(s): The reviewers have opted to remain anonymous.

Transaction Report:

DOI: <https://doi.org/10.1128/mSystems.00957-21>

October 13, 2021

Dr. Carmen Alicia Rivera Pérez
Georg-August-University Göttingen
Forest Botany and Tree Physiology
Göttingen
Germany

Re: mSystems00957-21 (Transcriptional Landscape of Ectomycorrhizal Fungi and Their Host Provide Insight into N Uptake from Forest Soil)

Dear Dr. Carmen Alicia Rivera Pérez:

Thank you for submitting your manuscript to mSystems. We have completed our review and I am pleased to inform you that, in principle, we expect to accept it for publication in mSystems. However, acceptance will not be final until you have adequately addressed the reviewer comments.

Preparing Revision Guidelines

Sincerely,

Emily Cope

Editor, mSystems

Journals Department
Reviewer comments:

Reviewer #1 (Comments for the Author):

This is a well written study with interesting findings. I only have a few comments.

Typo line 84, wrapps

There are too many acronyms which makes the paper hard to read. Please go over your acronyms and decide if the phrase is really used enough to justify using an acronym.

Acronym used before it is defined:

In the cyclic GS-GOGAT pathway, glutamine synthetase (GS)

98 catalyzes the formation of glutamine by transfer of ammonium to glutamate. Then

99 glutamate synthase (GOGAT)

You mention that you first used UNITE to identify your sequences. Using a database with a single taxonomic group leads to a high rate of false positives. I would use the nt database first, then UNITE, and compare the identifications you get from the 2 databases. Please see Marcelino et al. 2020. The use of taxon-specific reference databases compromises metagenomic classification. BMC Genomics

"Extrinsic

537 domain ASVs and unclassified ASVs were discarded from the taxonomic table." I think it would be better to have a line indicating how many were unclassified.

Why do you define both ASV and OTU? What is the difference between the 2 and what is the advantage of using both terms?

Line 546: need to define CTAB

"RNA reads were mapped against the reference transcriptomes of

559 *Fagus sylvatica* and 17 fungal species belonging to the same genera as those detected

560 by ITS barcoding

On average, 61 %

566 of the reads could be mapped (45 % to beech and 16 % to fungi)"

What do you think the other 40% is? Why didn't you use the NR database for your RNAseq data? Using such a small database to identify your sequences is problematic. Please see Marcelino et al. 2020. The use of taxon-specific reference databases compromises metagenomic classification. BMC Genomics

Please include an explanation of whether you sequenced your negative controls to account for reagent and lab contaminants. Or in other words, you removed the sequences from your negative controls from all other libraries. This would be for both barcoding and RNAseq analysis.

Reviewer #2 (Comments for the Author):

In this study, the authors assessed the role of ectomycorrhizal fungi in nitrogen uptake by beech trees. The study presented several novel findings including that the tree and the fungal community appear to respond to nitrogen surges independently from each other. Additionally, it was found that the fungal metatranscriptome was resistant to change with increased concentration of ammonia and nitrate.

Overall, I found the study to be engaging and well written aside from a few minor spelling errors. Additionally, I thought the analyses were appropriate. One of my only concerns with the study is the relatively low level of replication (n=4) per treatment. While this is understandable due to costs, especially for the RNAseq analysis, it is especially low for the amplicon analysis. However, I believe the study is still useful, this limitation should just be mentioned especially when showing fungal community composition. This data could also be better represented by showing the composition of each sample instead of merging in Figure 1 (see line by line comments).

My other major suggestion is that the full fungal metatranscriptomic dataset should be analyzed further in the same way that the EMF dataset was analyzed. This will be useful to see if there are any shifts in nitrogen metabolism in non EMF taxa. Again, I found this study to be very enjoyable and interesting. Further comments can be found in the line by line revisions file.

Line 17/18 - Is this the only way that N is acquired by these trees? If not, it should be stated that it may be acquired by the plants or other microbes etc.

Line 77- What is meant by this: "known discrimination between nitrate and ammonia"? Do some EMF only have the ability to take up one type? Please clarify.

Line 181-188- It may be worth representing these values as log₂FC instead of fold change, it can make them easier to interpret

Line 195 - Define the abbreviation EC numbers before using

Line 205 - It would be useful to look for these same functions in the full fungal dataset (including non-ectomycorrhizal) to see if there was any change in the nitrogen metabolism of non-EMF taxa

Line 221 - Why would the soil N content not be impacted by the 15N application while the root tips were? This may be worth mentioning in the discussion

Lines 292-294 - Again, I have some confusion about where these numbers of increased soil N come from since in line 221, it says that 15N application did not impact the soil

Line 309/310 - Consider giving the full name of this fungus instead of shortening to *C. geophilum* so it is clear what taxon is being discussed.

Line 367 - As stated above, it would be worth doing some of the same analyses on the non-EMF taxa to see if nitrogen metabolism was impacted

Line 466-468- Was there any difference in the cosms based on the day they were sampled? How many of the final 12 samples used for transcriptomics and community profiling were from each date? Also, were samples taken at the same time of day?

Line 532 and others - Since these were clustered at 97% they should just be called OTUs and not Amplicon Sequence Variants (ASVs). ASV implies single nucleotide differences and not clustering at 97%.

Line 565 - This mapping percentage seems very low. Is there any idea of what the remaining reads may have mapped to?

Line 607 - State how data was transformed

Figures and Tables:

Figure 1: It is stated that there was little significant difference between the treatments in terms of fungal community composition, but this does not seem to be the case for some taxa. In both the amplicon data and the transcript data, scleroderma is highly abundant in the control, but absent from the N treatments and Xereocomus is more abundant in the N treatments. Why might this be?

- It would be useful to show a bargraph representing all 12 samples instead of merging by treatment. This could show if differences were due to one sample or if the differences were consistent across treatments.

Figure 3: Give the full names of the Fungi in the legend

Response to Reviewer Comments

We would like to express our gratitude to the referees for taking the time to review our manuscript and for providing constructive comments and suggestions for its improvement. We have incorporated into the manuscript changes that reflect these valuable suggestions. Our point-by-point response (in blue) to the reviewers' comments and suggestions (in black), both in text and line-by-line, is provided below. Together with the changes, line numbers are provided for reference.

In addressing the comments and suggestions, in the revised manuscript, we corrected grammatical errors, limited the use of acronyms, implemented additional information regarding sequencing of controls, number of replicates, and details on data transformation. Additionally, we implemented a new figure (Fig. S2) showing the ITS2 metabarcoding and RNA seq results for each of the individual samples. We also added up-to-date results from KEGG pathway enrichment analysis for the full fungal metatranscriptome list since this was previously given for the ectomycorrhizal fungi metatranscriptome only.

Moreover, in the point-by-point response, we present the results from an alternative approach (i.e., ITS2 relative abundance of 23 samples, and mapping RNA seq results against the nt and nr database) for comparison with our original approach and provide an explanation of why our original approach is better suited for the goal of our study.

Reviewer #1 (Comments for the Author):

This is a well written study with interesting findings. I only have a few comments.

Thank you very much for your comments and suggestions. We highly appreciate your valuable feedback.

Comment: Typo line 84, wrapps

Thank you. We have corrected it to "enwraps" in Line 84.

Comment: There are too many acronyms which makes the paper hard to read. Please go over your acronyms and decide if the phrase is really used enough to justify using an acronym.

Acronym used before it is defined:

In the cyclic GS-GOGAT pathway, glutamine synthetase (GS) [Line 98] catalyzes the formation of glutamine by transfer of ammonium to glutamate. Then [Line 99] glutamate synthase (GOGAT)

Thanks for your suggestion. We have revised the manuscript to define acronyms before using them and limited their use if unnecessary. Enzymes have names commonly used by the community and when we refer to genes, the full names are spelled out, followed by symbols in parentheses to match nomenclature used in public databases.

Comment: You mention that you first used UNITE to identify your sequences. Using a database with a single taxonomic group leads to a high rate of false positives. I would use the nt database first, then UNITE, and compare the identifications you get from the 2 databases. Please see Marcelino et al. 2020. The use of taxon-specific reference databases compromises metagenomic classification. BMC Genomics

Thank you for your comment. We fully agree that using taxon-specific databases leads to false mapping of assembly-free metagenomic reads as it was nicely shown in Marcelino et al.

2020. However, we would like to emphasize that in contrast to the dataset referred in Marcelino et al. 2020, which was generated using shotgun metagenome sequencing (of all genomic contents in the environmental DNA sample), our data was generated from barcode/amplicon sequencing using fungal-specific primers.

Since our main interest was to target the ectomycorrhizal fungi dwelling on the beech root tips, we targeted the highly variable ITS2 rDNA region, which is currently the official barcode for species identification as it allows better discrimination in comparison to the more conserved small (18S) and large (28S) rRNA marker genes (Schoch et al., 2012; Nilsson et al., 2019a). ITS metabarcoding, in combination with Illumina Miseq technology, is a conventional and popular method used to study fungal community composition in environmental samples based on the relative abundance of either the fungal ITS1 or ITS2 region, with the latter having less taxonomic bias than the former (Nilsson et al., 2019a). The UNITE database, which contains all public fungal and other eukaryotic ITS sequences for the past 18 years and is subject to quality control measures including manual curation, is the formal reference database for taxonomic assignment of fungal ITS amplicon sequences (Nilsson et al. 2019b; <https://plutof.ut.ee/#/doi/10.15156/BIO/786373>). Therefore, the UNITE database is prioritized over the nt database.

Our pipeline further includes an additional blast step against the nt database after mapping against the high quality UNITE database so that the remaining “unclassified” sequences can be classified into fungal or non-fungal. These “unclassified” ITS amplicons are expected to belong to fungal sequences of unknown lineage, to be under pending curation in UNITE, or to be ITS sequences from non-fungal eukaryotes because despite using fungal-specific primers, the specificity is not always 100%. By mapping against the nt database, from 183 leftover “unclassified” sequences, we identified 181 of them as being fungal sequences and two extrinsic domains which were discarded from our data before proceeding with further analysis.

Nilsson, R. Henrik, Anslan, S., Bahram, M., Wurzbacher, C., Baldrian, P., & Tedersoo, L. (2019a). Mycobiome diversity: high-throughput sequencing and identification of fungi. *Nature Reviews Microbiology*, 17(2), 95–109. doi: 10.1038/s41579-018-0116-y

Nilsson, Rolf Henrik, Larsson, K. H., Taylor, A. F. S., Bengtsson-Palme, J., Jeppesen, T. S., Schigel, D., ... Abarenkov, K. (2019b). The UNITE database for molecular identification of fungi: Handling dark taxa and parallel taxonomic classifications. *Nucleic Acids Research*, 47(D1), D259–D264. doi: 10.1093/nar/gky1022

Schoch, C. L., Seifert, K. A., Huhndorf, S., Robert, V., Spouge, J. L., Levesque, C. a., ... Schindel, D. (2012). Nuclear ribosomal internal transcribed spacer (ITS) region as a universal DNA barcode marker for Fungi. *Proceedings of the National Academy of Sciences*, 109(16), 6241–6246. doi: 10.1073/pnas.1117018109

Comment: "Extrinsic [Line 537] domain ASVs and unclassified ASVs were discarded from the taxonomic table." I think it would be better to have a line indicating how many were unclassified.

Thanks for your suggestion. We have now added this information to the manuscript indicating that “Two non-fungal OTUs were discarded from the taxonomic table” in Line 559-560.

Comment: Why do you define both ASV and OTU? What is the difference between the 2 and what is the advantage of using both terms?

Thank you for bringing this point up. We have corrected in Line 554-556: “The raw sequences were quality filtered, merged, size filtered, denoised, and chimera checked. These high-quality sequences were clustered at 97% sequence identity into operation taxonomic units (OTUs) and abundance tables were generated.”

Explanation: ultimately, it comes down to how the used terminology is defined. By definition, Amplicon Sequence Variants allow distinction of sequence variation based on changes on a single nucleotide, whereas Operational Taxonomic Units result from clustering the sequences at a certain similarity threshold. In our pipeline, the sequences are first detected, quality filtered, and identified as ASVs since they are unique, then these “high-quality sequences” or ASVs are clustered according to 97% genetic identity (basically the definition of OTUs) because this clustering threshold is conventional for fungi. From the detection step to the clustering step, we keep the name as ASVs (e.g, ASV_0001) to be able to trace them back, but we could also call them ITS_0001 and the results would remain the same. Since we ultimately conduct community analyses on what is conventionally defined as OTUs, we now define only this term in the manuscript to avoid confusing the readers.

Comment: Line 546: need to define CTAB

Thank you, we have now defined “*hexadecyltrimethylammonium bromide*” in the manuscript (Line 574).

Comment: “RNA reads were mapped against the reference transcriptomes of [Line 559] *Fagus sylvatica* and 17 fungal species belonging to the same genera as those detected [Line 560] by ITS barcoding. On average, 61 % [Line 566] of the reads could be mapped (45 % to beech and 16 % to fungi)” What do you think the other 40% is?

Thank you for your question. Explanation: we did check this. The remaining reads belong to other unknown lineages, fungi of unknown lineage, non-ectomycorrhizal fungi like saprotrophs, endophytes, or pathogens, non-fungal eukaryotes, and to a minor extent, prokaryotes, and viruses. Since we extracted total RNA from beech ectomycorrhizas (i.e., symbiotic organs composed of ectomycorrhizal fungal cells and beech root cells) and used polyA enrichment for mRNA library preparation, most of the RNA should belong to European beech and ectomycorrhizal fungi and to a lesser extent to other fungi, other eukaryotes, and non-eukaryotic lineages. Contaminant rRNA was removed post-sequencing and comprised <2.31% in each sample.

Comment: Why didn't you use the NR database for your RNAseq data? Using such a small database to identify your sequences is problematic. Please see Marcelino et al. 2020. The use of taxon-specific reference databases compromises metagenomic classification. BMC Genomics

Thanks for your question. Explanation: currently, there are numerous bioinformatic tools available for metatranscriptomic data analyses because no single approach fits all needs and the choice depends on the aim of the experiment. When it comes to eukaryote metatranscriptomics, the biggest challenge is the limited availability of reference genomes, and this is especially true when it comes to studying the wide variety of fungal lineages with ectomycorrhizal lifestyle. As in any metatranscriptomics experiment, the taxonomic identity of the fungi present in the beech root tips of our experiment was unknown. Therefore, first we resolved the genus-level taxonomic

identity of the fungi through amplicon sequencing of the fungal ITS2 region with Illumina Miseq technology and by mapping the reads against the UNITE database and the nt database. Then, using FUNGUild, we determined whether the different fungal genera identified in the samples had either ectomycorrhizal, saprotrophic, endophytic or unknown lifestyle. Note that even though we targeted all types of fungi by ITS barcoding, we were mainly interested in targeting the symbiotic partners (ectomycorrhizal fungi and European beech). Thus, after obtaining genus and lifestyle information for the fungal genera detected in the samples, we checked whether the ectomycorrhizal fungi known to occur in the samples had been genome sequenced.

Initially, a number of ectomycorrhizal fungal genomes (*Amanita rubescens*, *Cortinarius glaucopus*, *Xerocomus badius*, *Boletus edulis*, *Russula ochroleuca*, *Lactarius quietus*, *Thelephora terrestris*) or saprotrophic fungal genomes (*Mycena galopus*) representing the genera detected in our samples were not publicly available in NCBI. Thus, if we would have just mapped the reads against nt we would have mapped to close neighbors available in the database. Note, however, in contrast to NCBI, the reference sequences and annotations of the relevant ectomycorrhizal fungi in our experiment were already accessible in the Joint Genome Institute (JGI) by the data owners. [The NCBI only releases the data after a specified release date, usually after a manuscript of the data is published, while JGI makes the data available earlier. Discussions to solve differences between the two databases are ongoing: <https://jgi.doe.gov/statement-on-the-use-of-genomics-data/>].

Since we did consider the possibility that mapping metatranscriptome reads against a limited number of reference sequences could lead to erroneous mapping, we made a compromise. Thus, we selectively included the reference sequences and annotations of the ectomycorrhizal fungi (downloadable from JGI) representing the genera known to occur in the experiment, the European beech (*Fagus sylvatica*) reference, as well as other non-ectomycorrhizal fungi (like saprotroph, endophytes, ericoid mycorrhizal fungi) also known to occur in our samples.

At a later point in the future, the reference sequences of the ectomycorrhizal fungi of interest became publicly available in NCBI after the data owners published (Miyauchi et al. 2020). Following this, we checked an alternative approach by assigning NCBI taxonomy to our metatranscriptomic data using a pipeline that implements both Kraken2 against the nt database and Kaiju against the nr database (both version: 13 July 2021). We found that the taxonomic assignment of the metatranscriptomic data using this approach is generally in agreement with our initial method since the majority of the classified fungal transcripts were assigned to the same fungal genera (and species) as those that we had previously determined to be important: ectomycorrhizal fungi (*Xerocomus*, *Amanita*, *Scleroderma*, *Lactarius quietus*, *Thelephora*, *Cenococcum*, *Russula*, *Boletus*, *Laccaria*, *Meliniomyces*, *Cortinarius*), saprotrophic fungi (*Mycena*, *Galerina*), ericoid mycorrhiza (*Oidiodendron*), endophyte (*Phialocephala*). Additionally, other non-ectomycorrhizal fungi present in the roots, non-fungal eukaryotes, and prokaryotes were identified. Please see the details below.

Comments: Please include an explanation of whether you sequenced your negative controls to account for reagent and lab contaminants. Or in other words, you removed the sequences from your negative controls from all other libraries. This would be for both barcoding and RNA-Seq analysis.

Thank you for your comment regarding this point. We have implemented this information in the barcoding and RNA seq sections of the manuscript:

Line 562-564: *“Initially, 23 samples and two controls (positive and negative) were sequenced; however, we did not find evidence of reagent contamination in the negative control and only the 12 samples for which RNA sequencing was done were included in further analyses.”*

Line 584-586: *“Since there was no amplification in the negative control of the final library PCR, the negative control was not sequenced.”*

Explanation:

We agree that this step is important. Generally, and to the best of our ability, we try to minimize sources of contamination; we worked with new and sterilized reagents/materials and carefully aliquoted reagents for use as necessary and have distinct workstations for specific steps of the molecular workflow. Nevertheless, and despite all precautions, we know that contamination is still possible. Therefore, we sequenced the negative controls that we processed together with the rest of the ITS barcoding samples. We found that contamination was negligible because in the negative control we only found one single read belonging to ASV_000013 (assigned to the Boletacea). In contrast, there were 115,346 read counts belonging to ASV_000013 in the experimental samples. Therefore, we did not subtract the single read count from the rest of the samples as we considered it to be negligible and could also have been the result of index hopping.

Similar precautions were taken in the RNA seq methodology, working with sterilized reagents and materials and using distinct workstations designated for different steps in the workflow. Moreover, our RNA extraction protocol includes a LiCl precipitation step which has high precipitation efficiency for RNA and low efficiency for protein and DNA, which is a potential contaminant in mRNA library preparation. To remove any leftover DNA from the RNA samples, we used the “Rigorous DNase treatment” protocol of the Turbo DNA-free kit which removes contaminant DNA to below PCR detection levels. Furthermore, the sequencing company notified us that there was no amplification in the negative control samples in the final library PCR. Although we are aware that minor contaminations at any step cannot be ruled out (e.g., the reagents loaded to the flow cell for sequencing could be theoretically contaminated, even though they should not), sequencing a negative control sample would require at least one whole lane in the flow cell, and based on results from previous RNA sequencing projects from our working group, sequencing the negative control was not necessary nor recommended by the sequencing company. In terms of the data, lineages with extremely low mapping percentage might be indicators of laboratory contaminants and thus, interpretation regarding those organisms should be done with caution.

Reviewer #2 (Comments for the Author):

In this study, the authors assessed the role of ectomycorrhizal fungi in nitrogen uptake by beech trees. The study presented several novel findings including that the tree and the fungal community appear to respond to nitrogen surges independently from each other. Additionally, it was found that the fungal metatranscriptome was resistant to change with increased concentration of ammonia and nitrate.

Comment: Overall, I found the study to be engaging and well written aside from a few minor spelling errors.

Thank you very much for your feedback. We are very grateful for your valuable comments and suggestions. We have corrected the spelling errors in our manuscript.

Comment: Additionally, I thought the analyses were appropriate. One of my only concerns with the study is the relatively low level of replication ($n=4$) per treatment. While this is understandable due to costs, especially for the RNA-Seq analysis, it is especially low for the amplicon analysis. However, I believe the study is still useful, this limitation should just be mentioned especially when showing fungal community composition. This data could also be better represented by showing the composition of each sample instead of merging in Figure 1 (see line by line comments).

Thank you for bringing these points up. Please see our explanations below:

RNA-Seq: We appreciate that the high costs associated with RNA-Seq, given the sequencing depth we used, did not go unnoticed. With our idea to study the response of natural assemblages of the ectomycorrhizal fungal communities living in association with European beech roots, we were walking into new territory where there was no clear consensus on the number of replicates that would be needed to detect differentially expressed genes. Since we previously tried a similar experiment with 4 replicates per treatment and 30 million reads per sample and found that the sequencing depth was not sufficient for capturing the fungi, in the present study we increased the sequencing depth to 100 million reads per sample while keeping four replicates per condition. However, even three replicates are still acceptable in RNA-Seq experiments (Conesa et al., 2016). Our approach and results were reassuring because we detected the expected biological responses to treatment in European beech and we were able to capture the fungi. The big challenge with field samples is the heterogenous nature of ectomycorrhizal fungal colonization of the fine roots. Nevertheless, our study provides a setting stone for similar field-studies in the future, when more ectomycorrhizal genomes become available, and large-scale approaches become more feasible.

ITS2: we agree that the costs for amplicon sequencing are reasonable. Originally, we sequenced 23 samples (Line 562). However, since our goal was to use the information from the fungal ITS2 amplicon sequencing as guide to select the reference fungal transcriptomes for mapping our RNA data, we included only the 12 samples for which we also had RNA-Seq data since we wanted to correlate ITS2 fungal results with RNA-Seq results from the same samples. Since naturally assembled ectomycorrhizas are quite heterogenous in the roots, adding more samples did not provide additional useful information.

In Figure 1, we want to emphasize the comparisons between the DNA and RNA results according to treatment. Also, our analysis was done at the metatranscriptome level and not at the individual fungi level. Nevertheless, we agree that it would be useful for readers to be able to see the complete picture for each of the samples. Therefore, we added Fig. S2 in the supplements showing the ITS2 barcoding and RNA seq results for each of the samples.

FIG S2 Heatmap of the relative abundances of European beech root-associated fungi based on ITS2 gene barcoding (A) and raw transcript counts of the metabolically active fungi based on RNA sequencing (B). The top 30 most abundant genera are shown in A. Transcript abundance for each of the 17 fungal species representing major ectomycorrhizal, saprotrophic, ericoid mycorrhizal, and endophytic fungi genera are shown in B. (Note: *Imleria badia* and *Xerocomus badius* are synonyms).

Comment: My other major suggestion is that the full fungal metatranscriptomic dataset should be analyzed further in the same way that the EMF dataset was analyzed. This will be useful to see if there are any shifts in nitrogen metabolism in non EMF taxa. Again, I found this study to be very enjoyable and interesting. Further comments can be found in the line by line revisions file.

Thanks for your comment. Our main goal was to study how the ectomycorrhizal fungi living in association with beech roots of forest-grown trees respond to shifts in nitrogen. Since other fungi like saprotrophs and endophytes are not directly involved in transferring nitrogen to the roots, and do not make structures in the fine roots, like the ectomycorrhizas that we harvested, we did not include a larger number of them as reference for mapping our transcriptomic data. However, we included three non-ectomycorrhizal fungi to lower the chances of erroneous mapping which could occur if we had only included ectomycorrhizal fungal references. Keeping this in mind, we did analyze the full fungal metatranscriptomic dataset and presented the results in the original manuscript. Perhaps this point was somehow overlooked during review.

We conducted differential expression analysis in both transcriptomic datasets: ectomycorrhizal fungi only and all fungi (Line 177-191, original manuscript). Then, as shown in Figure 3, the transporters and enzymes necessary for nitrate and ammonium uptake and assimilation are shown for the full fungal list in our dataset. We have now added KEGG enrichment analysis results for the full fungal metatranscriptome as well in Line 213-217 and Table 2.

Line Revisions File:

Comment: Line 17/18 - Is this the only way that N is acquired by these trees? If not, it should be stated that it may be acquired by the plants or other microbes etc.

Thank you for pointing this out. We have clarified this in Line 16-18: “Acquisition and translocation of N to forest trees is achieved *mainly* by highly diverse ectomycorrhizal fungi (EMF) living in symbioses with their host roots” because it is well known that, while EMF have a major role in N translocation to the roots, the trees can also acquire N directly.

Comment: Line 77- What is meant by this: “known discrimination between nitrate and ammonia”? Do some EMF only have the ability to take up one type? Please clarify.

Thanks for your suggestion. Based on studies conducted until now, ammonium uptake seems to be favored when both ammonium and nitrate are equally available, and while some EMF fungi have a limited number of nitrate transporters, it has been shown that many can use nitrate. However, this topic still needs further investigation, especially for ectomycorrhizal fungi in field conditions. Therefore, Line 76-78: “In general, ectomycorrhizal fungi have a preference for ammonium in comparison to nitrate (32, 33), but their ability to metabolize nitrate is also widespread (34, 35).”

Comment: Line 181-188- It may be worth representing these values as log2FC instead of fold change, it can make them easier to interpret

Thank you. We have corrected this in the manuscript (Line 190, 191, and 195).

Comment: Line 195 - Define the abbreviation EC numbers before using

Thank you. We have corrected this in the manuscript in Line 203 to “Enzyme Commission numbers.”

Comment: Line 205 - It would be useful to look for these same functions in the full fungal dataset (including non-ectomycorrhizal) to see if there was any change in the nitrogen metabolism of non-EMF Taxa

Yes, this is exactly what we did (Line 177-191, Line 204-211, and Fig. 3 of the original manuscript). In the revised manuscript: Line 183-196, Line 213-217, Line 217-225, and Fig. 3. Following your suggestion, we further added the results from KEGG pathway enrichment analysis for the full fungal metatranscriptome to the ectomycorrhizal one in Line 213-217 and Table 2.

Comment: Line 221 - Why would the soil N content not be impacted by the ^{15}N application while the root tips were? This may be worth mentioning in the discussion

Thanks for your question. The total nitrogen pool is composed of organic and inorganic N fractions. When ammonium or nitrate become available, plants and microbes including fungi and bacteria quickly take it up and metabolize it into organic N (plants) or other forms through microbial transformation. In our cosm system, the trees are the major living organism expected to take up the biggest fraction of the ^{15}N applied, either through the ectomycorrhizal fungal pathway or without EMF help. Another portion will be taken up by the ectomycorrhizal fungi (who need to acquire sufficient N to meet their own N demand as well as that of their host tree) and other soil-dwelling microbes including other types of fungi and bacteria. In the adjustment period, beech trees and microbes should have been a bit “hungry” by the time the N was applied since they were being watered with deionized water. Thus, when ammonium or nitrate was added to the soil, the trees and the microbes would “feed” on the easy-to-metabolize mineral source we applied. This is supported by our finding of higher ^{15}N contents in the ectomycorrhizal root tips (ectomycorrhizal fungal cells + beech root cells), followed by the fine roots and the coarse roots, while being depleted from the soil of ^{15}N -treated cosms. In the soil, very small amounts of the applied ^{15}N remained ($\sim 0.02 \text{ mg } ^{15}\text{N g}^{-1} \text{ soil}$) in excess of natural abundance in both ^{15}N ammonium- and ^{15}N nitrate-treated cosms. This very small amount made no significant difference to the total N pool (organic + inorganic N, which was about $\sim 4 \text{ mg N g}^{-1} \text{ soil}$) by the time we harvested. Also, important to mention is that N content in the soil is more heterogeneous and more difficult to measure than in ectomycorrhizas (fungi + root cells), fine and coarse root tissues. When mixing the soil for ^{15}N analyses, ^{15}N -rich and ^{15}N -poor microsites in the entire soil pool get homogenized and the aliquot that is measured gives a picture of the “average” of these two pools, making the value of the leftover ^{15}N in the soil, after fungal + root uptake, even smaller. Microbial activity also leads to the formation of N_2O which is released to the atmosphere. But these patterns of ^{15}N depletion in the soil compared to fungi and roots are well known in ^{15}N isotope studies. The main point in our study was that the ^{15}N source that we provided was traced and found in fungal and plant tissues after the incubation period that we allocated. We observed the typical ^{15}N uptake patterns in the ectomycorrhizas and in the roots, and depletion in the soil. Thus it was appropriate to sample at this time point and study their transcriptomes.

Comment: Lines 292-294 - Again, I have some confusion about where these numbers of increased soil N come from since in line 221, it says that ^{15}N application did not impact the soil

These numbers in the discussion (now Line 306-311) refer to the nitrogen concentration becoming higher since we applied ^{15}N as ammonium or as nitrate to the soil to make it available for uptake by the fungi and the tree roots (at the start of the experiment). Whereas in line 221 of the original manuscript (now Line 235), in the results, refer to the amount of N that remains at the end of the experiment after the two days of “incubation.” Since the fungi and plant have already taken up the N during the incubation time, the soil is depleted of it. Also, please note that the ^{15}N

we apply influences the soluble inorganic nitrogen pool in the soil, while the total N is made of organic N as well. Please also have a look at the previous comment with an explanation why most of the applied ^{15}N is found in the ectomycorrhizas and in the roots and depleted in the soil by the time we harvest.

Comment: Line 309/310 - Consider giving the full name of this fungus instead of shortening to *C. geophilum* so it is clear what taxon is being discussed.

Thanks for your suggestion. We have corrected this in **Line 324:** "*Cenococcum geophilum*."

Comment: Line 367 - As stated above, it would be worth doing some of the same analyses on the non-EMF taxa to see if nitrogen metabolism was impacted

Thank you for your comment. Yes, this is exactly what we did (Line 177-191, Line 204-2011, and Fig. 3 of the original manuscript). In the revised manuscript: **Line 183-196**, **Line 213-217**, **Line 217-225**, and **Fig. 3**. Following your suggestion, we further added the results from the KEGG pathway enrichment analysis for the full fungal metatranscriptome in **Line 213-217** and in **Table 2**. Moreover, we have reworded the sentence (from Line 367 in the original manuscript and now **Line 386-387**) from "all EMF" to "all studied fungi" since we analyzed both EMF and all fungi in our list.

Comment: Line 466-468- Was there any difference in the cosms based on the day they were sampled?

Thanks for your question. To make these points clear in the manuscript, we added the following statements:

Line 507-508: "*For ^{15}N analyses, freeze-dried aliquots of soil, root tips, fine and coarse root samples from both experimental batches were milled using a ball mill...*").

Line 526-528: "*For determination of NH_4^+ , NO_3^- and non-structural carbohydrates, frozen fine roots (-80°C), 12 samples ($n = 4$ per treatment) were milled (MM400, Retsch GmbH) under liquid nitrogen...*"

Line 562-564: "*Initially, 23 samples and two controls (positive and negative) were sequenced; however, we did not find evidence of reagent contamination in the negative control and only the 12 samples for which RNA sequencing was done were included in further analyses.*"

Line 572-574: "*Total RNA was isolated from 25 of the frozen powder beech root tip samples using an extraction method based on hexadecyltrimethylammonium bromide (117).*"

Line 576-579: "*Twelve samples with RNA integrity ranging from 6.7 to 7.9 were selected for polyA selection and mRNA library preparation (Table S6). These samples also have the corresponding ITS2 metabarcoding sequencing data and are all from the same experimental batch.*"

Explanation: initially, 23 samples and controls were sequenced for fungal community profiling (ITS2 metabarcoding). However, for RNA sequencing we selected 12 samples with the best RIN values and these samples were all from the same experimental batch (batch 2). Since our goal was to use the ITS2 metabarcoding results as guide for selecting the reference fungal transcriptomes for mapping our RNA-Seq data and given that fungal composition in forest-grown trees is quite heterogenous, we analyzed the 12 barcoded samples for which we also had

sequenced the RNA. Our goal was to correlate the ITS2 gene relative abundance with the transcript abundance of the fungal genera (Fig 1). Therefore, since we only analyzed samples from the same batch, between batch comparisons do not apply. We do not expect to see an effect of treatment on tree biomass, but biomass info is provided to inform whether there was variation in the different conditions since the trees were randomly assigned to treatment. While ¹⁵N analyses was done in samples from both batches, sometimes a compartment is missing because there was not sufficient material, therefore batch comparisons are not possible. Ammonium-N, nitrate-N and carbohydrates measurements were done in fine roots of the 12 samples that were also used for DNA and RNA sequencing (all from the same batch). In the manuscript we provide sample size information for each of the respective analyzes, which should clarify this. We also provide all the raw data in Supporting Data set 6.

Comment: How many of the final 12 samples used for transcriptomics and community profiling were from each date? Also, were samples taken at the same time of day?

Thanks for asking. All samples for both metabarcoding and RNA seq derived from the same batch. We have clarified this now (please see the answer above). We further state in Line 490-491: “*The cosms were harvested in the morning 48 h after initial ¹⁵N application in alternating order according to treatment.*”

Comment: Line 532 and others - Since these were clustered at 97% they should just be called OTUs and not Amplicon Sequence Variants (ASVs). ASV implies single nucleotide differences and not clustering at 97%.

Thank for suggesting this point. We have corrected in the manuscript in Line 554-556: “*The raw sequences were quality filtered, merged, size filtered, denoised, and chimera checked. These high-quality sequences were clustered at 97% sequence identity into operation taxonomic units (OTUs), and abundance tables were generated.*”

Comment: Line 565 - This mapping percentage seems very low. Is there any idea of what the remaining reads may have mapped to?

Thanks for the question. Considering that mapping was done against 17 representative fungal species and European beech, while in reality the samples were composed of more fungi, other microeukaryotes, etc., this percentage is actually quite good because it covers the major ectomycorrhizal fungi detected in the samples by the ITS barcoding approach. To put it into perspective in terms of mapping against organisms from the same genus, only 87% of the reads resulting from whole transcriptome sequencing of total RNA isolated from leaves of *Populus x euramericana* could be mapped to the reference genome of *Populus deltoides* (Ning et al., 2019). Moreover, one of the main challenges is the patchy nature of the ectomycorrhizal fungi presence in roots of field trees (some genera are present in some, but not in other samples, therefore, mapping was heterogenous within replicates of the same condition. Therefore, our approach was to aggregate the fungi according to KOGs into a metatranscriptome. In this way the read coverage is more uniform and suitable for differential analyses.

We also tested an alternative approach. We assigned NCBI taxonomy to our metatranscriptomic reads using both the nt and nr databases using a combination of Kraken2 (prioritized) and Kaiju, respectively (both version: 13 July 2021). We found that the taxonomic assignment is generally in agreement with our initial method since the majority of the classified fungal transcripts were assigned to the same fungal species as we originally determined to be important: ectomycorrhizal fungi (*Xerocomus*, *Amanita*, *Scleroderma*, *Lactarius*, *Thelephora*,

Line 643: “For cluster analysis of N-related transporters and enzymes for all the fungi in the metatranscriptomic dataset (Fig 3), the original transcript values were $\ln(x + 1)$ -transformed.”

Line 646: “Details on data transformation is indicated in the respective figure or table legends.”

Comment: Figures and Tables:

Figure 1: It is stated that there was little significant difference between the treatments in terms of fungal community composition, but this does not seem to be the case for some taxa. In both the amplicon data and the transcript data, scleroderma is highly abundant in the control, but absent from the N treatments and Xereocomus is more abundant in the N treatments. Why might this be? - It would be useful to show a bar graph representing all 12 samples instead of merging by treatment. This could show if differences were due to one sample or if the differences were consistent across treatments.

Thank you for bringing this point up. A new figure showing all 12 samples for both the barcoding and the RNA-based results is now provided in the supplements (Fig. S2, please also find it above). In Figure 1, we emphasize the comparisons between the ITS2 gene relative abundance results for the fungal genera and correlate them to the metatranscriptomic reads to the relevant fungal genera in the samples. It is not our goal to emphasize in this figure the within-treatment variations resulting from the well-known heterogeneous nature of ectomycorrhizal fungi in the roots of forest trees. The statistical test showed “no significant differences” in fungal community composition at the OTU level (the proxy for species-level) with $P = 0.861$. Nevertheless, we agree that it would be useful for the reader to detailed sample information in Fig. S2, where it can be observed that although fungi like *Xerocomus* and *Cenococcum*, occur almost in all samples, the number of reads is too low for differential expression analyses. This was also a reason why we did not conduct differential expression analyses on individual fungi.

Comment: Figure 3: Give the full names of the Fungi in the legend

Thanks for suggesting this idea. We have now written the full names in the legend.

November 29, 2021

Dr. Carmen Alicia Rivera Pérez
Georg-August-University Göttingen
Forest Botany and Tree Physiology
Göttingen
Germany

Re: mSystems00957-21R1 (Transcriptional Landscape of Ectomycorrhizal Fungi and Their Host Provide Insight into N Uptake from Forest Soil)

Dear Dr. Carmen Alicia Rivera Pérez:

Your manuscript has been accepted, and I am forwarding it to the ASM Journals Department for publication. For your reference, ASM Journals' address is given below. Before it can be scheduled for publication, your manuscript will be checked by the mSystems senior production editor, Ellie Ghatineh, to make sure that all elements meet the technical requirements for publication. She will contact you if anything needs to be revised before copyediting and production can begin. Otherwise, you will be notified when your proofs are ready to be viewed. Please address Reviewer one's final comments at this stage.

Publication Fees:

We recognize that the video files can become quite large, and so to avoid quality loss ASM suggests sending the video file via <https://www.wetransfer.com/>. When you have a final version of the video and the still ready to share, please send it to Ellie Ghatineh at eghatineh@asmusa.org.

Sincerely,

Emily Cope
Editor, mSystems

Journals Department
Fig. S2: Accept
Table S6: Accept
Text S1: Accept
Table S2: Accept
Table S4: Accept
Table S5: Accept
Table S1: Accept
Fig. S1: Accept
Table S3: Accept